# Midbrain glutamatergic circuit mechanism of resilience to socially transferred allodynia in male mice

Yi Han [1,2,3,10], Lin Ai[1,2,3,10], Lingzhen Song[1,2,3,10], Yu Zhou[1,2,3,10], Dandan Chen[1,2,3], Sha Sha[1,2,3], Ran Ji[1,2,3], Qize Li[1,2,3], Qingyang Bu[1,2,3], Xiangyu Pan[1,2,3], Xiaojing Zhai[1,2,3], Mengqiao Cui[1,2,3], Jiawen Duan[4], Junxia Yang[1,2,3], Dipesh Chaudhury [5], Ankang Hu[6], He Liu [7], Ming-Hu Han [4,8,11] ✉, Jun-Li Cao [1,2,3,9,11] ✉ & Hongxing Zhang [1,2,3,11] ✉

The potential brain mechanism underlying resilience to socially transferred allodynia remains unknown. Here, we utilize a well-established socially transferred allodynia paradigm to segregate male mice into pain-susceptible and pain-resilient subgroups. Brain screening results show that ventral tegmental area glutamatergic neurons are selectively activated in pain-resilient mice as compared to control and pain-susceptible mice. Chemogenetic manipulations demonstrate that activation and inhibition of ventral tegmental area glutamatergic neurons bi-directionally regulate resilience to socially transferred allodynia. Moreover, ventral tegmental area glutamatergic neurons that project specifically to the nucleus accumbens shell and lateral habenula regulate the development and maintenance of the pain-resilient phenotype, respectively. Together, we establish an approach to explore individual variations in pain response and identify ventral tegmental area glutamatergic neurons and related downstream circuits as critical targets for resilience to socially transferred allodynia and the development of conceptually innovative analgesics.

Clinical studies have demonstrated that certain individuals report severe sensory discomfort (susceptible to pain), whereas others report experience less pain (resilient to pain) under the condition of similar noxious insults and tissue damage[1–8]. Such individual variations in pain response are well documented in both health and disease contexts[9,10]. For instance, rare outlier individuals among patients with inherited erythromelalgia (IEM) report experiencing less pain, whereas the majority of them suffer from intolerable intense pain[1,11–15]. While most studies in pain research have focused on the mechanisms of pain susceptibility and related pathology[1], recent investigations have started to unveil the peripheral mechanisms of pain resilience[11–15]. Given the complexity of the pain process, involving pain detection and

[1]Jiangsu Province Key Laboratory of Anesthesiology, Xuzhou Medical University, Xuzhou, Jiangsu 221004, PR China. [2]Jiangsu Province Key Laboratory of Anesthesia and Analgesia Application Technology, Xuzhou Medical University, Xuzhou, Jiangsu 221004, PR China. [3]NMPA Key Laboratory for Research and Evaluation of Narcotic and Psychotropic Drugs, Xuzhou Medical University, Xuzhou, Jiangsu 221004, PR China. [4]Department of Mental Health and Public Health, Faculty of Life and Health Sciences, Shenzhen Institute of Advanced Technology, Chinese Academy of Sciences, Shenzhen, Guangdong 518055, PR China. [5]Division of Science, New York University Abu Dhabi (NYUAD), Saadiyat Island 129188, United Arab Emirates. [6]The Animal Facility of Xuzhou Medical University, Xuzhou Medical University, Xuzhou, Jiangsu 221004, PR China. [7]Department of Anesthesiology, Huzhou Central Hospital, Huzhou, Zhejiang 313000, PR China. [8]Department of Pharmacological Sciences, Icahn School of Medicine at Mount Sinai, New York, NY 10029, USA. [9]Department of Anesthesiology, The Affiliated Hospital of Xuzhou Medical University, Xuzhou, Jiangsu 221004, PR China. [10]These authors contributed equally: Yi Han, Lin Ai, Lingzhen Song, Yu Zhou. [11]These authors jointly supervised this work: Ming-Hu Han, Jun-Li Cao, Hongxing Zhang. ✉e-mail: hanmh@siat.ac.cn; caojl0310@aliyun.com; hongxing.zhang@xzhmu.edu.cn

perception in the peripheral and central nervous systems, respectively, it is striking that the brain mechanism underlying pain resilience still remains unknown.

The interest and attention on pain and stress resilience have dramatically increased in recent decades. For example, KCNQ2 encoding a subtype of inhibitory potassium (K+) channels in the dorsal root ganglia and ventral tegmental area (VTA) has been identified as a target for pain resilience and social stress resilience[11–16], which leads to the potential therapeutic utility of KCNQ channel opener, retigabine (also called ezogabine), as a potential analgesic and antidepressant[17–21]. Functional magnetic resonance imaging studies have demonstrated substantial disparities in gray matter volume and functional connectivity within the default mode network underlying subjective reports of experiencing more or less resilient response to pain states[22–24], indicating both anatomical and functional variations in the brain of resilient individuals. These studies on resilience open a different avenue to develop conceptually innovative therapies for major depressive disorder and pain[1,11–16,21–23,25–30].

However, the mechanisms through which the resilient brain copes with pain experience are poorly understood. This is partially attributable to the lack of a reliable experimental paradigm that permits stable and consistent replication of inter-individual differences in pain response in laboratory animals. The establishment of severe tissue damage or inflammation makes it an obstacle to recapitulate susceptible versus resilient phenotypes in current animal pain models, such as those induced by complete Freund's adjuvant (CFA) to elicit persistent inflammatory pain and the sciatic nerve ligation-induced neuropathic pain[31–34]. Recent animal studies have demonstrated that the state of hyperalgesia could be socially transferred from one individual to another through a brief empathetic social contact, a pain model referred to as socially transferred pain[35–39]. Given the evident inter-individual variation in empathy processing and social behaviors[28,40,41], this model provides an opportunity to identify different pain profiles.

In this work, we utilize a standard protocol for socially transferred allodynia (STA), also known as empathic pain[42], to segregate STA mice into pain-susceptible and pain-resilient sub-populations by combined use of two cutoff criteria: one based on mechanical paw withdrawal thresholds (PWTs) after empathetic social contact, and the other based on the ratio of PWTs after versus before empathetic social contact. Our whole-brain c-Fos protein immunostaining demonstrates an elevated activity of glutamatergic neurons in the VTA of STA resilient mice, a well-known brain region involved in mediating pain and social behaviors[25–30,34]. Moreover, we take advantage of cell-type-specific chemogenetic approaches and systematically investigated the role of the VTA glutamatergic neurons in mediating resilience to pain. Through broader screening, we also identify the functions of two downstream neural circuits associated with these neurons in the development and maintenance of resilience to STA.

## Results

### Susceptible and resilient signatures post-STA
Utilizing the well-established mouse paradigm of socially transferred pain (Fig. 1a)[35,42], we first observed that following a 1-h brief social contact with a familiar male cage mate experiencing CFA inflammatory pain, male C57BL/6J bystander (BY) mice displayed a remarkable decrease in PWTs, a measurement of mechanical allodynia examined with von Frey test (Fig. 1a). Our results show that these behavioral profiles could last at least 6 h and returned to the baseline level 24 h after the social contact (Supplementary Fig. 1). By analyzing a substantial number of BY mice from multiple experiments, we observed a wide bimodal distribution of PWTs responses: when examined immediately after social contact, ~70% of BY mice displayed decreased PWTs, while the remaining ~30% exhibited comparable PWTs with the control mice (Fig. 1b). According to the K-

means clustering analysis, PWTs value of 0.41 was set as a cutoff for sub-population segregation: BY mice with PWTs ≤ 0.41 were labeled as susceptible to STA (BY-S), and those with PWTs > 0.41 were labeled as the resilient sub-population (BY-R) (Fig. 1c). The resilient sub-population had median and variance values comparable to their controls (Supplementary Table 1). In addition to the PWTs value, PWTs ratio was used to define the relative pain response, which is equal to the ratio of PWTs after versus before the brief social contact. Our data revealed a similar bimodal distribution of PWTs ratios (Fig. 1d). K-means clustering analysis identified a ratio of 75% as the cutoff, which effectively distinguishing the susceptible and resilient sub-populations at proportions similar to those observed with the PWTs cutoff of 0.41 (Fig. 1e and Supplementary Table 2). Consistently, >90% of BY-S and BY-R animals identified using the two different cutoffs overlapped (191/209 mice, Fig. 1f). The proportion of each sub-population was not influenced by prolonging the social contact to 2 h (Fig. 1g). To further test the stability of this inter-individual difference in pain responses, we replicated the modeling process 1–2 weeks after the initial social contact. Similarly, >90% of mice (101/112 mice) displayed the same behavioral phenotype as observed after the initial modeling process (Fig. 1h). And these behavioral changes were consistently observed in both hind paws (Fig. 1i). Interestingly, female C57BL/6J mice displayed similar susceptible and resilient phenotypes following the 1-h social contact process with a similar proportion to the male mice (Supplementary Fig. 2 and Supplementary Tables 3 and 4). These results underscore the stability of the STA paradigm in recapitulating the inter-individual differences in pain responses in mice.

Moreover, our behavioral results during and after the STA paradigm revealed similar social interaction, allogrooming and targeted allolicking behaviors in both BY-S and BY-R mice (Supplementary Figs. 3 and 4), indicating the inter-individual difference in pain responses between the two sub-populations was not derived from the difference in social interactive behaviors.

### Activation of VTA^Glu neurons by resilience
Next, we looked for the brain regions specifically involving in STA resilience. To do so, we performed immunofluorescent staining with the c-Fos protein antibody in the brain of mice after the brief social contact and von Frey tests (Fig. 2a, b). Quantitative data showed an increase of c-Fos protein level in most of the tested brain regions, including those that were well-established in BY mice, such as the anterior cingulate cortex (ACC), nucleus accumbens (NAc) and periaqueductal gray (PAG) (Fig. 2c)[35]. The majority of brain regions showed comparable c-Fos protein expression level between the BY-S and BY-R sub-populations (Fig. 2c. ACC: Control versus BY-S, $P = 0.0078$; Control versus BY-R, $P = 0.4076$; BY-S versus BY-R, $P = 0.3004$. NAc: Control versus BY-S, $P = 0.0005$; Control versus BY-R, $P = 0.3700$; BY-S versus BY-R, $P = 0.0087$. NAc core: Control versus BY-S, $P = 0.0001$; Control versus BY-R, $P = 0.4640$; BY-S versus BY-R, $P = 0.0019$. NAc shell: Control versus BY-S, $P = 0.0286$; Control versus BY-R, $P = 0.9307$; BY-S versus BY-R, $P = 0.0944$. Paraventricular thalamic nucleus: Control versus BY-S, $P = 0.0126$; Control versus BY-R, $P = 0.0015$; BY-S versus BY-R, $P = 0.8046$. Basolateral amygdala: Control versus BY-S, $P = 0.0100$; Control versus BY-R, $P = 0.9998$; BY-S versus BY-R, $P = 0.0100$. Central amygdala: Control versus BY-S, $P = 0.0079$; Control versus BY-R, $P > 0.9999$; BY-S versus BY-R, $P = 0.0006$. Lateral habenula: Control versus BY-S, $P = 0.0695$; Control versus BY-R, $P = 0.0310$; BY-S versus BY-R, $P = 0.9680$. Reuniens thalamic nucleus: Control versus BY-S, $P = 2.5 \times 10^{-5}$; Control versus BY-R, $P = 0.0034$; BY-S versus BY-R, $P > 0.9999$. Lateral hypothalamus: Control versus BY-S, $P = 0.0011$; Control versus BY-R, $P = 0.0538$; BY-S versus BY-R, $P = 0.8511$. PAG: Control versus BY-S, $P = 0.0314$; Control versus BY-R, $P = 0.9995$; BY-S versus BY-R, $P = 0.0404$. Dorsal raphe nucleus: Control versus BY-S, $P = 0.0004$; Control versus BY-R, $P = 0.8490$; BY-S

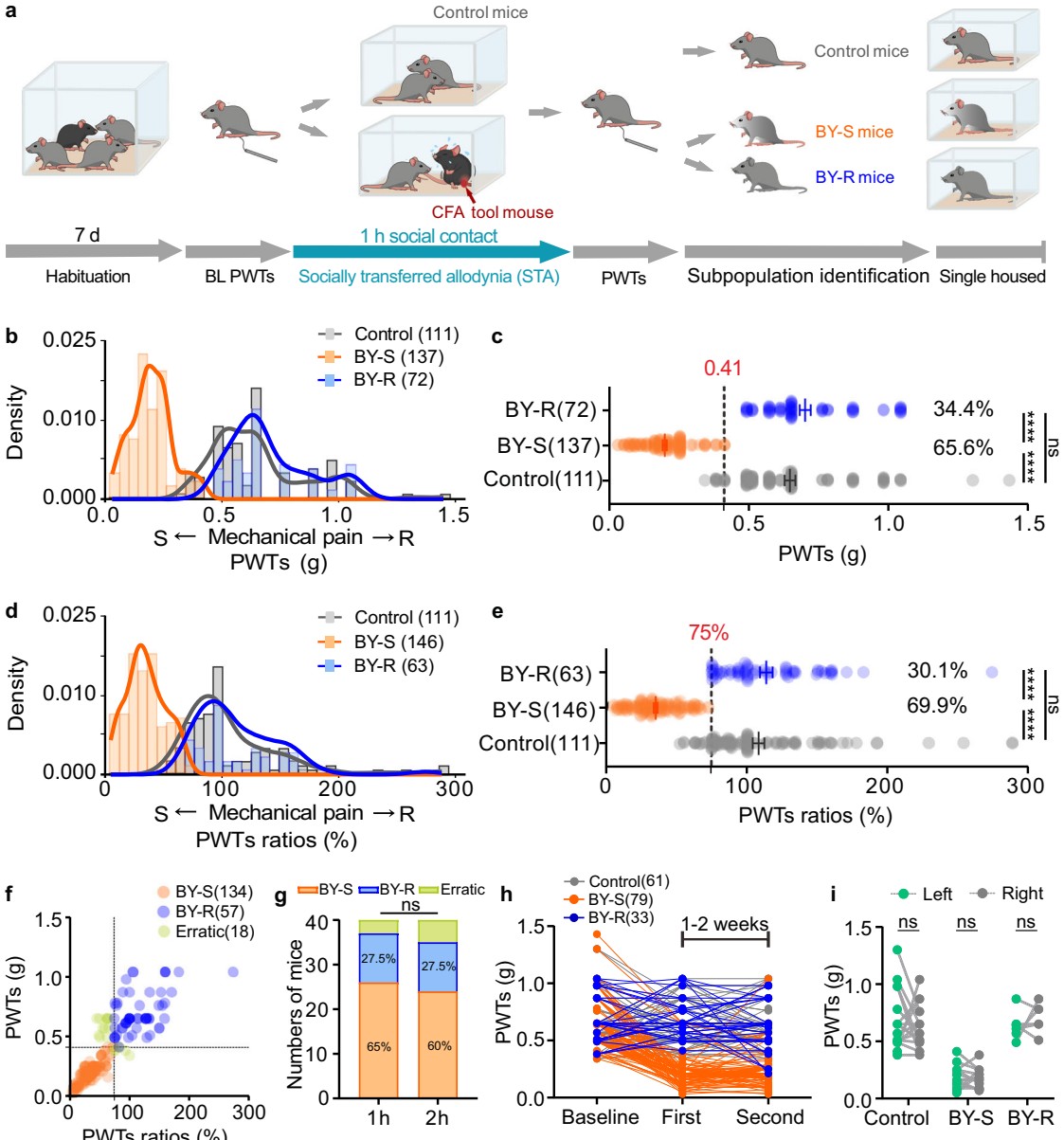

**Fig. 1 | Identification of susceptible and resilient sub-populations.**
**a** Experimental timeline. **b** PWTs distribution in control, susceptible and resilient mice ($n$ = 111, 137 and 72 mice). **c** Horizontal scatterplot depicting the distribution of PWTs ($n$ = 111, 137 and 72 mice). Control versus BY-S, $P$ < 0.0001; Control versus BY-R, $P$ = 0.4058; BY-S versus BY-R, $P$ < 0.0001. The black dashed line represents the threshold value dividing the two subgroups. **d** PWTs ratios distribution in control, BY-S and BY-R mice ($n$ = 111, 146 and 63 mice). **e** Horizontal scatterplot depicting the distribution of individual PWTs ratios ($n$ = 111, 146 and 63 mice). Control versus BY-S, $P$ < 0.0001; Control versus BY-R, $P$ = 0.9796; BY-S versus BY-R, $P$ < 0.0001. The black dashed line represents the threshold value dividing the two subgroups. **f** Scatterplot showing overlapped susceptible and resilient mice identified with the two methods ($n$ = 134, 57 and 18 mice). **g** Percentages of susceptible

and resilient mice after 1 h ($n$ = 26, 11 and 3 mice; left) and 2-h social contact ($n$ = 24, 11 and 5 mice; right). $P$ = 0.7483. **h** Individual PWTs plotted across two STA repeats at a 1-week or 2-week interval ($n$ = 61, 79 and 33 mice). **i** PWTs of the ipsilateral versus contralateral hind paws of the BY mice ($n$ = 18, 13 and 6 mice). BL baseline; PWTs paw withdraw thresholds; CFA complete Freund's adjuvant; BY-S the susceptible sub-population to socially transferred allodynia; BY-R the resilient sub-population to socially transferred allodynia; STA socially transferred allodynia. The data are presented as the mean ± s.e.m. ****$P$ < 0.0001, ns no significance. Data analyzed by (**c**, **e**) Kruskal–Wallis test with Dunn's multiple comparisons test; **g** chi-square test; or **i** two-way repeated-measures (RM) ANOVA with Tukey's multiple comparisons test. Statistical details are presented in Supplementary Table 5. Source data are provided as a Source Data file.

versus BY-R, $P$ = 4.1 × 10$^{-5}$. Locus coeruleus (LC): Control versus BY-S, $P$ = 3.9 × 10$^{-6}$; Control versus BY-R, $P$ = 0.0104; BY-S versus BY-R, $P$ = 0.1778). Interestingly, the c-Fos protein expression level in the VTA specifically increased only in the BY-R mice (Fig. 2c–e). Further immunohistochemistry experiments performed in C57BL/6J mice with tyrosine hydroxylase (TH) staining and *Vgat-IRES-Cre* or *Vglut2-IRES-Cre* mice with local expression of Cre-inducible rAAV-DIO-EGFP showed that the VTA glutamatergic neurons, but not GABAergic or dopaminergic neurons, were significantly activated in the BY-R mice

(Fig. 2f–k and Supplementary Fig. 5). To further confirm the hyperactivity of VTA glutamatergic neurons in BY-R mice, we labeled these neurons with Cre-inducible AAV carrying mCherry in *Vglut2-IRES-Cre* mice for slice electrophysiology. Cell-attached recordings demonstrated an increased firing frequency of VTA mCherry-positive neurons in BY-R mice when compared with that in the control or BY-S groups (Supplementary Fig. 6). These data suggest that VTA glutamatergic neurons may be involved explicitly in resilience to STA in the brain.

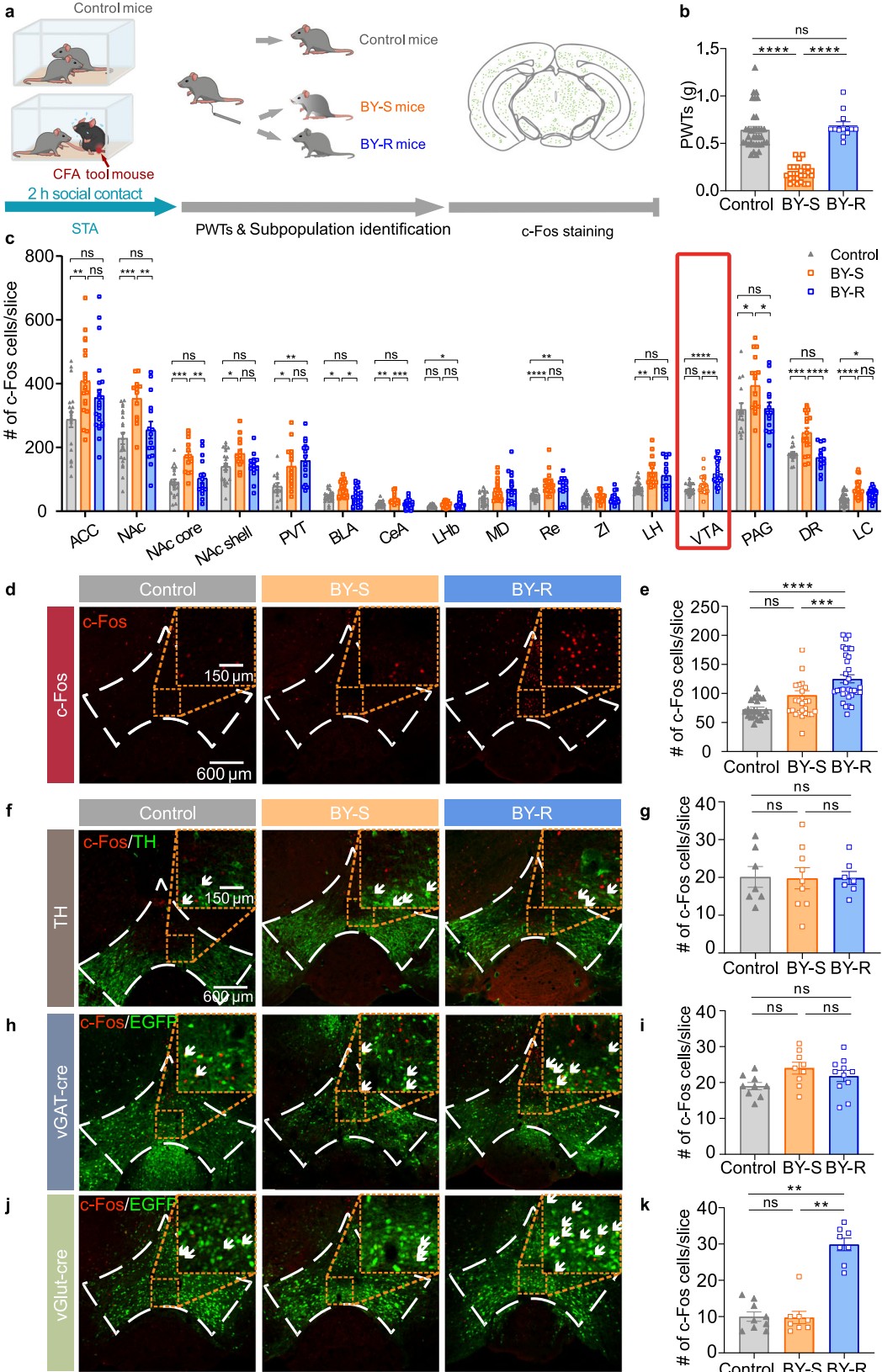

## Pro-resilience by activating VTA^Glu neurons

To test if the hyperactivity of VTA glutamatergic neurons would be sufficient to promote the development of resilience, we first injected rAAV2/9-EF1α-DIO-hM3D (Gq) into the VTA of *Vglut2-IRES-Cre* mice to chemogenetically activate these neurons (Fig. 3a). Immunohistochemistry staining conducted 2 h after CNO injection demonstrated an

increased expression of c-Fos protein level in VTA glutamatergic neurons (Fig. 3b, c). Since the susceptible and resilient phenotypes were stable and recoverable (Fig. 1), we designed a 3-repeat modeling experiment at a 1-week interval to examine the effect of VTA glutamatergic neurons activation on the development of resilience to STA (Fig. 3a). Consistent with the data stated above, following the first

**Fig. 2 | Activation of VTA glutamatergic neurons in the resilient mice.**
**a** Experimental timeline (adapted from *The Mouse Brain in Stereotaxic Coordinates* by Paxinos and Franklin). **b** PWTs ($n$ = 35, 26 and 12 mice). Control versus BY-S, $P = 8.2 \times 10^{-10}$; BY-S versus BY-R, $P = 4.9 \times 10^{-8}$. **c** Quantification of c-Fos protein expression in different brain regions ($n$ = 15–35 slices from 5–9 mice). **d, e** Representative immunofluorescent images and quantitative data of VTA c-Fos protein expression ($n$ = 22, 25 and 33 slices from 7, 8 and 9 mice); Control versus BY-R, $P = 2.5 \times 10^{-7}$; BY-S versus BY-R, $P = 0.0007$. **f, g** Representative immunofluorescent images and quantitative data of c-Fos protein expression in VTA dopaminergic neurons ($n$ = 9, 8 and 8 slices from 3, 3 and 3 mice); Control versus BY-R, $P = 0.9998$; BY-S versus BY-R, $P > 0.9999$. **h, i** Representative immunofluorescent images and quantitative data of c-Fos protein expression in VTA GABAergic neurons ($n$ = 7, 9 and 7 slices from 3, 3 and 3 mice). Control versus BY-R, $P = 0.3702$; BY-S versus BY-R, $P = 0.5458$. **j, k** Representative immunofluorescent

images and quantitative data of c-Fos protein expression in VTA glutamatergic neurons ($n$ = 9, 9 and 11 slices from 3, 3 and 3 mice). Control versus BY-R, $P = 0.0017$; BY-S versus BY-R, $P = 0.0015$. White arrows indicate overlapped neurons. The staining was repeated twice with similar results. Scale bar: 600 and 150 μm. ACC anterior cingulate cortex; NAc nucleus accumbens; PVT paraventricular thalamic nucleus; BLA basolateral amygdala; CeA central amygdala; LHb lateral habenula; MD mediodorsal thalamic nucleus; Re reuniens thalamic nucleus; ZI zona incerta; LH lateral hypothalamus; VTA ventral tegmental area; PAG periaqueductal gray; DR dorsal raphe nucleus; LC locus coeruleus; TH tyrosine hydroxylase. Data are presented as the mean ± s.e.m. Data analyzed by (**b, e, g**) Kruskal–Wallis test with Dunn's multiple comparisons test, or (**i, k**) one-way ANOVA with Tukey's multiple comparisons test. ns no significance, *$P < 0.05$, **$P < 0.01$, ***$P < 0.001$, ****$P < 0.0001$. Statistical details are presented in Supplementary Table 5. Source data are provided as a Source Data file.

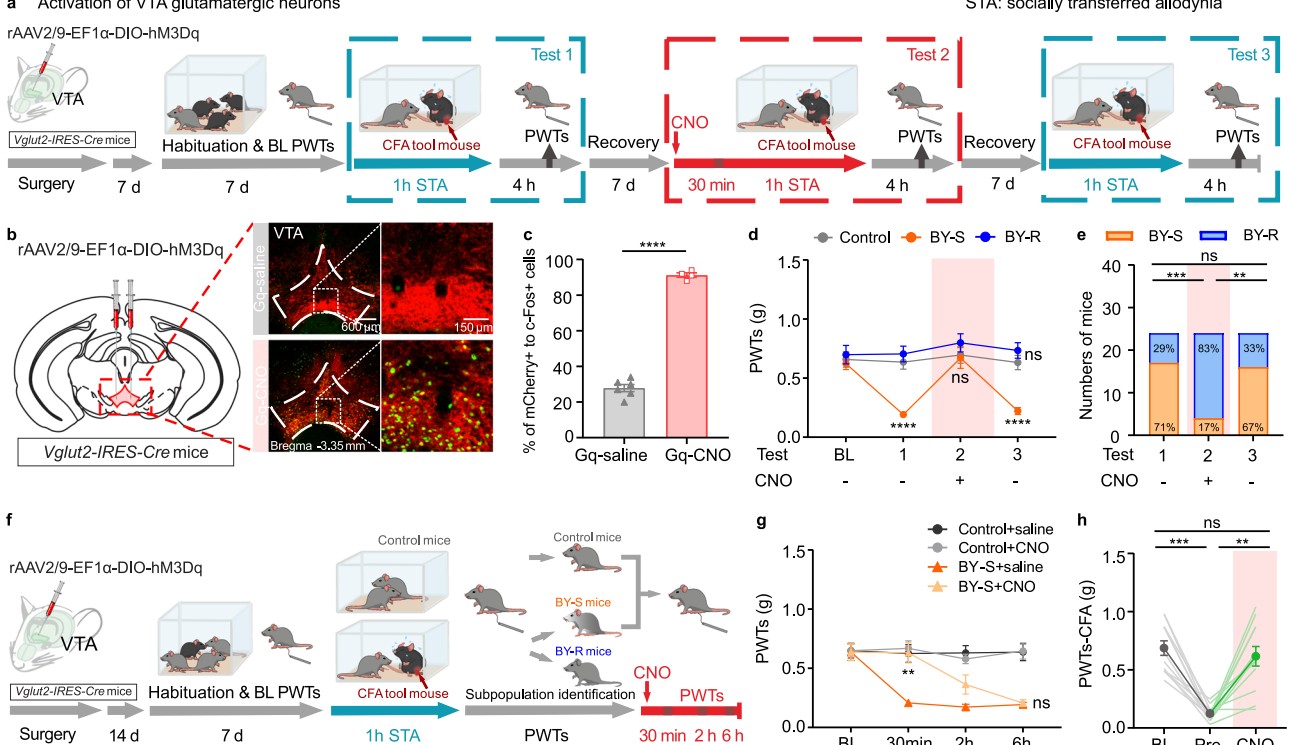

**Fig. 3 | Activation of VTA glutamatergic neurons promotes STA resilience.**
**a** Experimental timeline. **b, c** Representative immunofluorescent images and quantitative data of c-Fos protein expression in mCherry-positive VTA glutamatergic neurons after saline or CNO treatment in *Vglut2-IRES-Cre* mice (adapted from *The Mouse Brain in Stereotaxic Coordinates* by Paxinos and Franklin). Scale bar: 600 and 150 μm; $n$ = 6, 4 slices from 3 and 2 mice, $P = 1.6 \times 10^{-7}$. The staining was repeated twice with similar results. **d** PWTs for different subgroups of BY mice across three repeats of social transfer of allodynia paradigm with or without chemogenetic activation of VTA glutamatergic neurons, subgroups of BY mice were identified according to the behavioral outcome after their first exposure to social contact with CFA partners ($n$ = 13, 17 and 7 mice). Control versus BY-S, $P_{BL} = 0.8890$, $P_{T1} = 1.4 \times 10^{-5}$, $P_{T2} = 0.9706$, $P_{T3} = 2.9 \times 10^{-5}$; Control versus BY-R, $P_{BL} = 0.9152$, $P_{T1} = 0.7226$, $P_{T2} = 0.5950$, $P_{T3} = 0.5041$. **e** Percentages of BY-S and BY-R mice across

the three repeats of paradigm. Test 1 versus Test 2, $P = 0.0004$; Test 1 versus Test 3, $P > 0.9999$; Test 2 versus Test 3, $P = 0.0010$. **f** Experimental timeline. **g** PWTs over different time points before and after CNO injection in BY-S mice ($n$ = 10 mice per group). BY-S + saline versus BY-S + CNO, $P_{BL} > 0.9999$, $P_{30min} = 0.0034$, $P_{2h} = 0.1688$ and $P_{6h} = 0.9816$. **h** Individual and summary data showing the PWTs for CFA mice at baseline, before, and after CNO injection ($n$ = 12 mice). BL versus Pre, $P = 0.0005$; Pre versus CNO, $P > 0.9999$; BL versus CNO, $P = 0.0033$. The data are expressed as the mean ± s.e.m. **$P < 0.01$, ***$P < 0.001$, ****$P < 0.0001$, ns no significance. Data analyzed by (**c**) unpaired two-sided *t*-test; (**d, g**) two-way RM ANOVA with Tukey's post hoc test; (**e**) two-sided Fisher's exact test and (**h**) two-sided Friedman test with Dunn's multiple comparisons test. Statistical details are presented in Supplementary Table 5. Source data are provided as a Source Data file.

modeling session (Test 1), BY mice were divided into the BY-S subgroup with decreased PWTs and the BY-R subgroup with unchanged PWTs (BY-R:BY-S = 29%:71%, determined with a PWTs ratio of 75%, Fig. 3d, e). Thirty minutes before the second social contact session, all the BY mice received an intraperitoneal administration of CNO to activate the Gq-expressing VTA glutamatergic neurons (Test 2). Von Frey behavioral tests carried out 4 h post social contact (to wash out the possible real-

time analgesic effects of chemogenetic activation) showed a significant increase of PWTs in those previously identified BY-S mice and an augmented number of total BY-R mice (BY-R:BY-S = 83%:17%, Fig. 3d, e). As expected, the third modeling experiment without CNO administration (Test 3) yielded data similar to the first repeat (BY-R:BY-S = 33%:67%, Fig. 3d, e). A potent analgesic effect was also observed in the well-established BY-S mice upon CNO injection to activate their VTA

glutamatergic neurons, lasting <2 h (Fig. 3f, g). In contrast, no effects were observed in the control mice receiving CNO injection (Fig. 3g), indicating that activation of VTA glutamatergic neurons does not affect baseline PWTs. These results suggest a context-dependent pain-regulating effect by VTA glutamatergic neurons activation. Moreover, a robust analgesic effect was also observed in CFA mice following CNO treatment (Fig. 3h). Together, these data support that chemogenetic activation of VTA glutamatergic neurons is sufficient to promote the development of resilience during social contact and holds a potent analgesic property on established STA and inflammatory pain.

**Pro-susceptibility by inhibiting VTA^Glu neuron**
To evaluate the necessity of VTA glutamatergic neurons in determining the development of resilience to STA, we injected rAAV2/9-

EF1α-DIO-hM4D (Gi) into the VTA of *Vglut2-IRES-Cre* mice to chemogenetically inhibit these neurons (Fig. 4a). Immunohistochemistry confirmed the inhibitory effect of the chemogenetic approach (Fig. 4b, c). Similarly, following the first modeling session (Test 1), 65% of the BY mice were identified as susceptible with reduced PWTs, and the remaining 35% were resilient with relatively unchanged PWTs (Fig. 4d, e). Chemogenetic inhibition of the VTA glutamatergic neurons during the second modeling session (Test 2) induced a remarkable reduction of PWTs in mice (von Frey behavioral tests were carried out 4 h post social contact) that were previously identified as BY-R (Fig. 4d, e). Following the third modeling session (Test 3) without CNO administration, behavioral responses of the test mice recovered to the levels observed in Test 1 (BY-R:BY-S = 31%:69%, Fig. 4d, e). Interestingly, a pro-allodynia effect was also

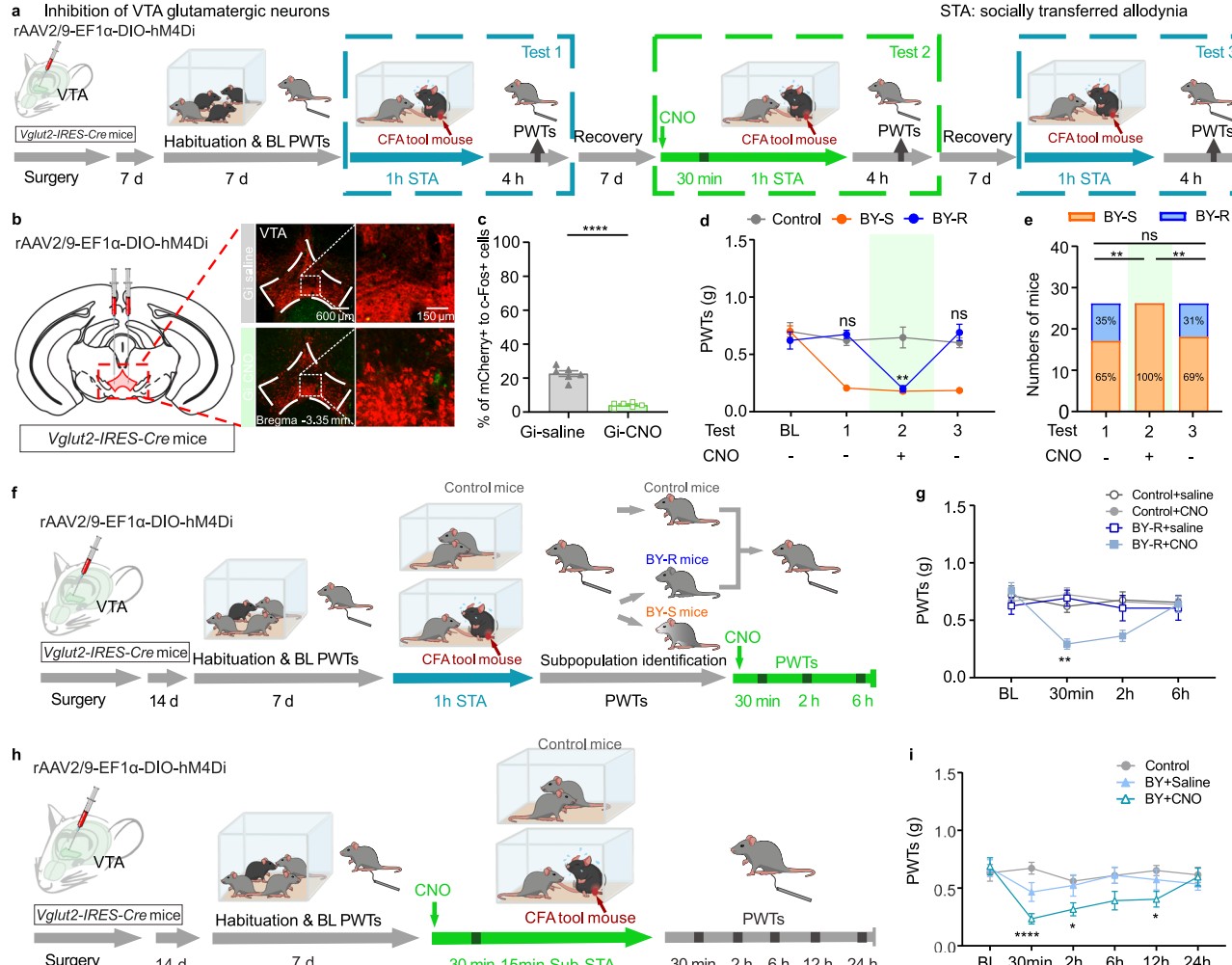

**Fig. 4 | Inhibition of VTA glutamatergic neurons promotes susceptibility to STA. a** Experimental timeline. **b**, **c** Representative immunofluorescent images and quantitative data of c-Fos protein expression in mCherry-positive VTA glutamatergic neurons after saline or CNO treatment in *Vglut2-IRES-Cre* mice (adapted from *The Mouse Brain in Stereotaxic Coordinates* by Paxinos and Franklin). Scale bar: 600 and 150 μm; $n = 6$, 6 slices from 3 mice/group, $P = 1.1 \times 10^{-6}$. **d** PWTs for different subgroups of BY mice across three repeats of social transfer of allodynia paradigm with or without chemogenetic inhibition of VTA glutamatergic neurons, subgroups of BY mice were identified according to the behavioral outcome after their first exposure to social contact with CFA partners ($n = 10$, 17 and 19 mice). Control versus BY-R, $P_{BL} = 0.7498$, $P_{T1} = 0.6023$, $P_{T2} = 0.0018$, $P_{T3} = 0.5510$. **e** Percentages of BY-S and BY-R mice across the three repeats of social transfer of allodynia paradigm. Test 1 versus Test 2, $P = 0.0017$; Test 1 versus Test 3, $P > 0.9999$; Test 2 versus Test 3, $P = 0.0042$. **f** Experimental timeline and chemogenetic inhibition of VTA

glutamatergic neurons in BY-R mice following the 1 h of social contact. **g** PWTs over different time points before and after CNO injection in BY-R mice ($n = 9, 13, 9$ and 8 mice). BY-R + saline versus BY-R + CNO, $P_{BL} = 0.5942$, $P_{30min} = 0.0017$, $P_{2h} = 0.2435$, $P_{6h} = 0.9935$. **h** Experimental timeline and chemogenetic inhibition of VTA glutamatergic neurons in BY mice during a 15 min of sub-threshold social contact. **i** PWTs after chemogenetic inhibition of VTA glutamatergic neurons during the sub-threshold social contact ($n = 9, 8$ and 8 mice). Control versus BY + CNO, $P_{BL} = 0.9082$, $P_{30min} = 2.2 \times 10^{-5}$, $P_{2h} = 0.0324$, $P_{6h} = 0.0647$, $P_{12h} = 0.0284$, $P_{24h} = 0.9961$. The data are presented as the mean ± s.e.m. *$P < 0.05$, **$P < 0.01$, ****$P < 0.0001$, ns no significance. Data analyzed by (**c**) unpaired two-sided *t*-test; (**d**, **g**, **i**) two-way RM ANOVA with Tukey's multiple comparisons test and (**e**) two-sided Fisher's exact test. Statistical details are presented in Supplementary Table 5. Source data are provided as a Source Data file.

observed in the established BY-R mice but not in the control mice upon receiving CNO injection, which could last about 2 h (Fig. 4f, g). To further explore the role of VTA glutamatergic neurons in the development of susceptibility, we implemented a 15-min sub-threshold paradigm, which was insufficient to induce behavioral changes. Thirty minutes before being subjected to the sub-threshold paradigm, mice expressing Gi in their VTA glutamatergic neurons received either CNO or saline treatment (Fig. 4h). As anticipated, chemogenetic inhibition of these neurons was sufficient to induce a significant decrease of PWTs in CNO-treated mice compared to those receiving saline, with a behavioral outcome lasting about 12 h (Fig. 4i). These results suggest that inhibiting VTA glutamatergic neurons exert a susceptibility- and allodynia-promoting effect in a context-dependent manner. Together with the chemogenetic activation studies, our data indicate that VTA glutamatergic neurons are a specific cellular target for resilience to STA.

## VTA^DA neuron modulates established STA

Dopamine neurons are the primary neuronal type in the VTA and have been implicated in social behavior and pain modulation[26–28,43]. Nevertheless, our chemogenetic experiments showed that bi-directionally regulating VTA dopamine neurons did not affect the development of

susceptibility or resilience to STA (Supplementary Figs. 7 and 8). On the other hand, chemogenetic activation of VTA dopamine neurons generated a significant analgesic effect in established pain states in CFA mice (Supplementary Fig. 7d).

## VTA^Glu circuits differently regulate STA

Next, we explored how VTA glutamatergic neurons regulate resilience to STA at the circuitry level. First, a Cre-inducible rAAV virus encoded with EGFP was micro-injected into the VTA of *Vglut2-IRES-Cre* mice to map the downstream brain regions that received VTA glutamatergic afferents. Whole-brain tracing results indicated a dense neuronal EGFP expression in VTA with a similar pattern as reported before (Fig. 5a, b)[44]. Fiber expression was observed in some of the well-known VTA downstream brain regions (Supplementary Fig. 9), including those implicated in STA, such as the NAc shell and lateral habenula (LHb) (Fig. 5c, d). Several subregions in the thalamus, hippocampus and brain stem also expressed relatively dense staining of EGFP (Supplementary Fig. 9).

The VTA has been well demonstrated to interact with the NAc and LHb in regulating social behavior-related psychiatric disorders and pathological pain behaviors[26,27,29,34,43,45,46]. We, therefore, hypothesized that these two projection circuits might control inter-individual

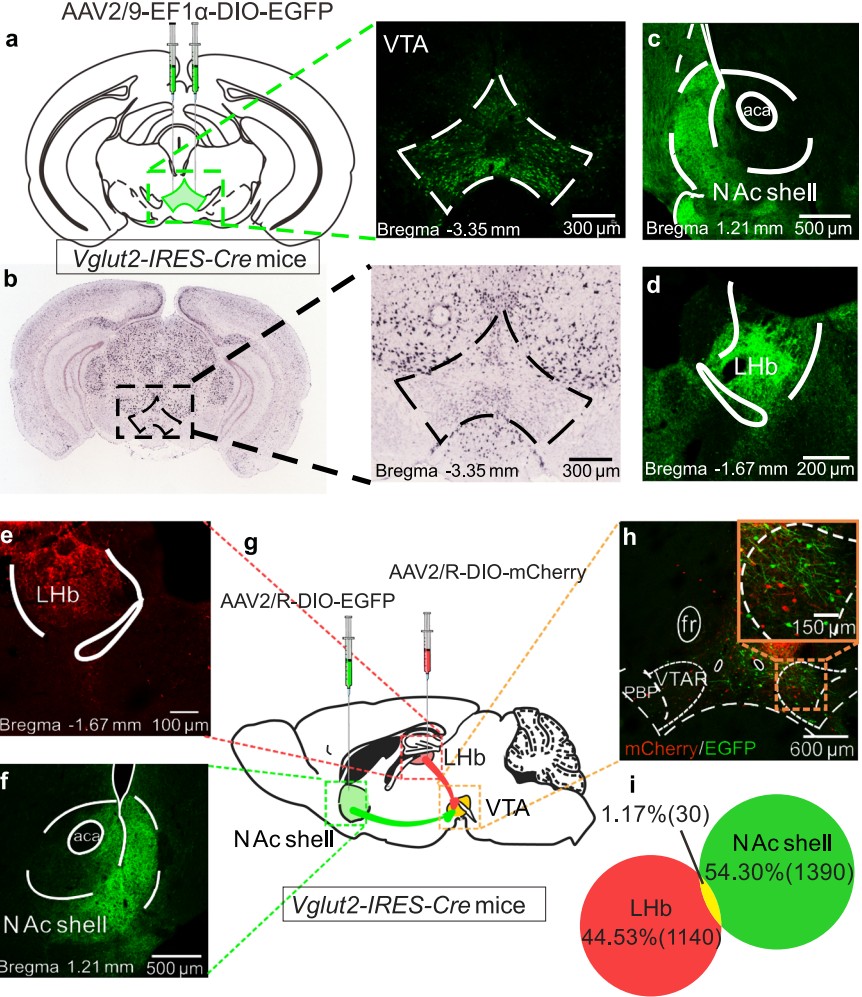

**Fig. 5 | VTA glutamatergic neurons that project to the NAc shell and LHb are from different sub-populations. a** Viral injection schematic and a representative immunofluorescent image showing EGFP-positive VTA neurons (adapted from *The Mouse Brain in Stereotaxic Coordinates* by Paxinos and Franklin). **b** A representative in situ hybridization image showing VTA *Slc17a6* gene (*Vglut2*) expression (adapted from Allen Brain Institute ISH data). **c, d** EGFP-positive fibers in the NAc

shell and LHb. **e–g** Schematic for retrograde tracing strategy, representative immunofluorescent images for viral injection sites in the NAc Shell and LHb. **h, i** Representative immunofluorescent images and quantitative data of VTA→NAc shell and VTA→LHb projecting glutamatergic neurons. Scale bars are shown in related images. The staining was repeated twice with similar results.

differences in STA. Retrograde labeling with Cre-inducible retrograde rAAV2/R encoding EGFP and mCherry in *Vglut2-IRES-Cre* mice revealed distinct sub-populations of VTA glutamatergic neurons projecting to the NAc shell and LHb with minimal overlap (Fig. 5e, i). To manipulate the VTA→NAc shell or VTA→LHb glutamatergic projections, FLP-dependent viruses rAAV2/9-EF1a-fDIO-hM3D(Gq)-EGFP-WPRE-hGH polyA or rAAV2/9-EF1a-fDIO-hM4D(Gi)-EGFP-WPRE-hGH polyA was

delivered into the VTA, and rAAV2/R-EF1a-DIO-FLP-WPRE-hGH polyA was micro-injected into the NAc shell or LHb of *Vglut2-IRES-Cre* mice (Fig. 6a). Behavioral tests found that chemogenetic activation of the VTA→NAc shell glutamatergic projection during the modeling process promoted resilient behavioral phenotype and generated more BY-R mice, whereas pathway-specific inhibition of the same circuit prevented the development of resilience, rendering all BY mice

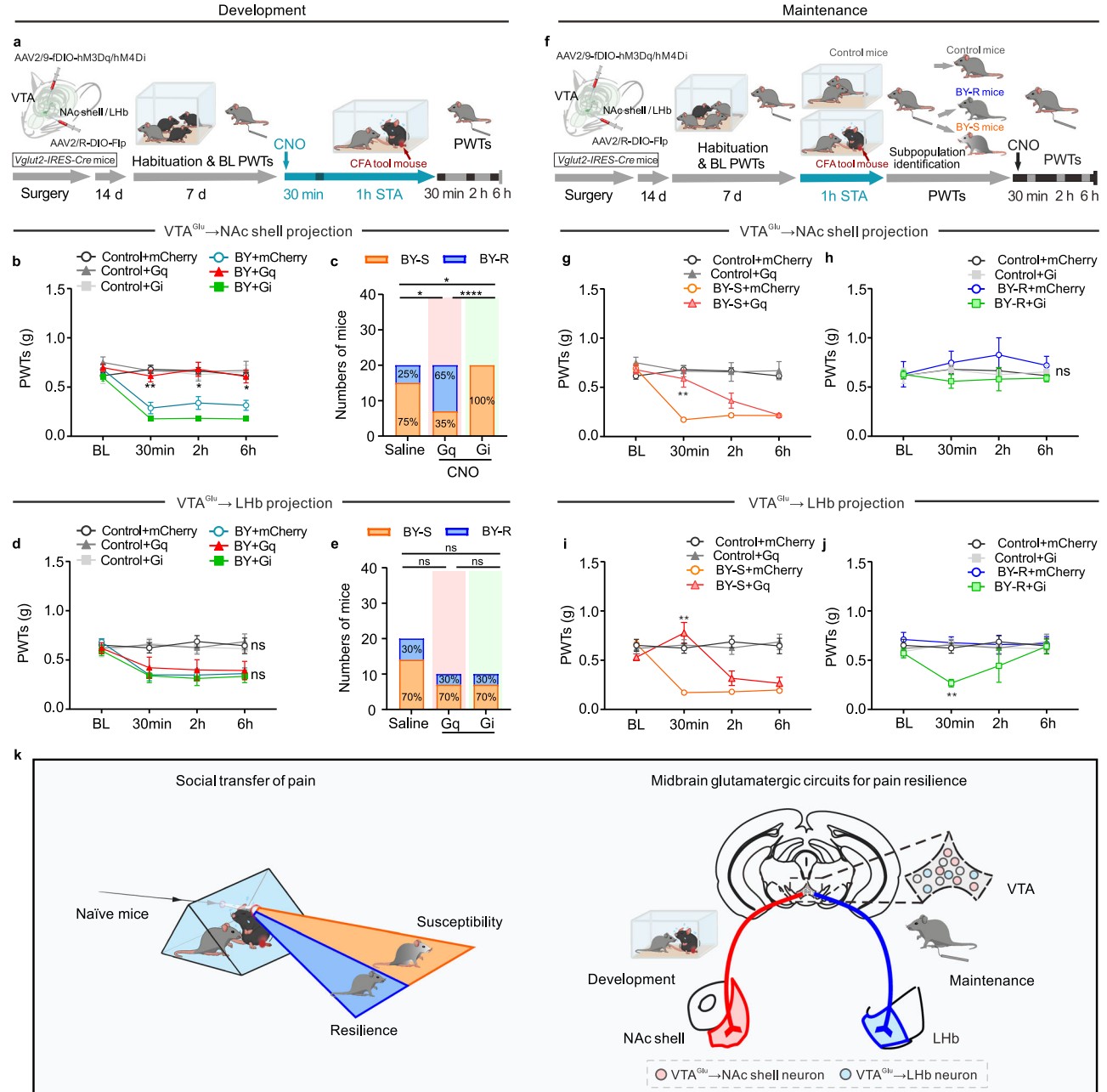

**Fig. 6 | VTA glutamatergic circuits distinctly regulate the development and maintenance of STA resilience. a** Experimental timeline. PWTs and sub-population percentages in BY mice with bidirectional manipulation of the VTA→NAc shell glutamatergic circuit during STA ($n$ = 20, 10, 7, 20, 20 and 16 mice). **b** BY + mCherry versus BY + Gq, $P_{30min}$ = 0.0044, $P_{2h}$ = 0.0110, $P_{6h}$ = 0.0115; BY + mCherry versus BY + Gi, $P_{30min}$ = 0.4665, $P_{2h}$ = 0.2358, $P_{6h}$ = 0.1384. **c** Saline versus Gq, $P$ = 0.0248; Saline versus Gi, $P$ = 0.0471; Gq versus Gi, $P$ = 1.3 × 10$^{-5}$. **d, e** PWTs and sub-population percentage in BY mice with bidirectional manipulation of the VTA→LHb glutamatergic circuit ($n$ = 10, 9, 9, 20, 9 and 10 mice). **f** Experimental timeline. **g, h** PWTs with bidirectional manipulation of the VTA→NAc shell glutamatergic circuit in BY-S (**g**, $n$ = 20, 10, 16 and 15 mice; BY + mCherry

versus BY + Gq, $P_{30min}$ = 0.0013) and BY-R (**h**, $n$ = 20, 7, 4 and 5 mice), respectively. **i, j** PWTs with bidirectional manipulation of the VTA→LHb shell glutamatergic circuit in BY-S (**i**, $n$ = 10, 9, 13 and 6 mice; BY + mCherry versus BY + Gq, $P_{30min}$ = 0.0077) and BY-R (**j**, $n$ = 10, 9, 7 and 3 mice; BY + mCherry versus BY + Gi, $P_{30min}$ = 0.0019) mice, respectively. **k** Summary diagram (adapted from *The Mouse Brain in Stereotaxic Coordinates* by Paxinos and Franklin). Data are expressed as the mean ± s.e.m. *$P$ < 0.05, **$P$ < 0.01, ****$P$ < 0.0001, ns no significance. Data analyzed by (**b, d, g, h, i, j**) two-sided two-way RM ANOVA with Tukey's multiple comparisons test, or (**c, e**) two-sided Fisher's exact test. Statistical details are presented in Supplementary Table 5. Source data are provided as a Source Data file.

susceptible to STA (Fig. 6b, c and Supplementary Fig. 10). Interestingly, bi-directional chemogenetic manipulation of the VTA→LHb glutamatergic projection had no impacts on the development of inter-individual behavioral difference (Fig. 6d, e and Supplementary Fig. 11). Furthermore, chemogenetic experiments in mice subjected to the sub-threshold paradigm further indicated that inhibition of the VTA→NAc shell glutamatergic projection, but not the VTA→LHb glutamatergic pathway, promoted the development of STA (Supplementary Fig. 12). These results suggest that VTA→NAc shell glutamatergic projection is a specific neural pathway for the development of resilience.

We then evaluated the functional role of these two glutamatergic projections in regulating established susceptible and resilient behavioral phenotypes. Chemogenetic experiments showed that activation of the VTA→NAc shell glutamatergic projection provoked resilience-like behavioral outcomes in previously identified BY-S mice (Fig. 6f). However, inhibition of this projection failed to reverse the established resilient behavior in BY-S mice (Fig. 6g, h). Surprisingly, chemogenetic manipulation of the VTA→LHb glutamatergic projection bi-directionally regulated the established behavioral susceptibility and resilience (Fig. 6i, j). In addition, behavioral tests in CFA mice demonstrated a robust increase of PWTs upon chemogenetic activation of VTA→LHb glutamatergic projection, but not the VTA→NAc shell glutamatergic circuit (Supplementary Fig. 13). These data indicate that the VTA→LHb glutamatergic pathway played a critical role in maintaining the established behavioral phenotypes.

Our results demonstrated that VTA glutamatergic neurons projecting to the NAc shell and LHb distinctly mediate the development and maintenance of inter-individual differences in socially transferred pain (Fig. 6k).

## Discussion

Here, we showed that mice subjected to a brief social contact with a painful cage mate could be reliably classified into the susceptible and resilient sub-populations using the von Frey pain test. Pain-resilient mice exhibited an elevated glutamatergic neuronal activity in the VTA, and chemogenetic manipulation of these neurons bi-directionally regulated the susceptible and resilient behavioral phenotypes to pain. Our projection-specific functional studies demonstrated that the VTA glutamatergic neurons distinctly governed the development and maintenance of pain by targeting different downstream brain regions, namely, the NAc shell and LHb.

Social contact with CFA mice caused increased neuronal activity in various brain regions across the rostro-caudal neural axis, as reported in a recent study[35]. Among these brain regions, we found that VTA, a well-known brain region for social contact, pain processing, and affective disorders[25–30,34,43], exhibited an increased neuronal activity specifically in the BY-R mice. Chemogenetic activation of the VTA glutamatergic neurons during social contact resulted in increased PWTs, which persisted for at least 6 h, providing a viable time window for further functional manipulation. Interestingly, chemogenetic activation of these neurons in BY-S mice provided ~2 h of alleviation from established STA. This line of evidence indicates that activation of VTA glutamatergic neurons during the modeling processing was sufficient to promote the development of resilient phenotype. Consistently, chemogenetic inhibition of these neurons hindered the development of resilience to STA and facilitated pain susceptibility in mice subjected to a shorter duration of social contact with CFA cage mates. These cell-type-specific studies established an essential role for VTA glutamatergic neurons in regulating resilience to STA.

It is widely recognized that VTA dopamine neurons project to a broad range of downstream brain regions, by which they modulate emotional and sensory states[25–30,34,43]. In contrast, the functional role of VTA glutamatergic projections has been relatively understudied. Our comprehensive whole-brain tracing results screening revealed

several downstream brain regions receiving glutamatergic inputs from the VTA, including the NAc shell, LHb and many other nuclei along the rostro-caudal axis of the brain. Recent research has highlighted the NAc core as a critical brain structure to confer STA in mice[35]: activation of the ACC glutamatergic afferents in the NAc core during the social contact period was sufficient to prolong STA, whereas inhibition of this projection reversed the established STA. Interestingly, our findings demonstrate chemogenetic manipulation in the VTA→NAc shell glutamatergic projection, but not the VTA→LHb glutamatergic projection, during the social contact bi-directionally regulated the development of susceptibility and resilience to STA. The divergence between these studies indicated a functional heterogeneity of NAc sub-nuclei and their excitatory afferents in regulating animal behaviors. Surprisingly, the VTA→LHb glutamatergic projection plays a more essential role in mediating well-established STA and CFA inflammatory pain. Specifically, the inhibiting activity of VTA→LHb glutamatergic projection reversed the established resilience to STA in BY-R mice, whereas increasing activity in this pathway provoked an analgesic effect in BY-S mice. In contrast, inhibiting VTA→NAc shell glutamatergic projection failed to regulate the established resilience to STA, albeit with an analgesic property when exciting the pathway. These results highlight the projection-specific regulatory role of the VTA glutamatergic neurons in shaping the development and maintenance of inter-individual differences in socially transferred pain.

VTA is a core brain region characterized by a high-density dopaminergic neuron (~60%) and a relatively small proportion of glutamatergic neurons (~5%)[47,48]. While both cell types have been implicated in mediating motivation-related behaviors, such as reward and aversion[49–52], the dopaminergic neurons were reported more for their particular functional roles in mediating stress-related mental disorders[26–29,53]. Notably, the VTA glutamatergic and dopaminergic neurons exhibit heterogeneity in their synaptic inputs, output and neurobiological functions[54]. The striatal regions and the pallidum contribute a more substantial portion of inputs to VTA dopaminergic neurons, whereas glutamatergic neurons predominantly receive cortical inputs[55]. In rodents, the major targets of VTA dopamine efferents are the medium spiny neurons in the NAc, with additional projections to the amygdala, cortex, hippocampus, ventral pallidum, PAG, bed nucleus of the stria terminalis, olfactory tubercle and LC[54]. Conversely, VTA glutamatergic neurons mainly project to the NAc, LHb, ventral pallidum and amygdala[56,57]. These anatomical differences might contribute to the functional disparities in mediating behaviors. Our retrograde tracing experiments in *Vglut2-IRES-Cre* mice showed that VTA glutamatergic neurons projecting to the NAc and LHb also express the dopaminergic neuron marker TH (Supplementary Fig. 14), suggesting that these neurons may simultaneously co-release dopamine from their axon terminals in the NAc and LHb. Our electrophysiological studies further revealed that optogenetic activation of the axon terminals in the NAc or LHb induced excitatory postsynaptic currents in their respective postsynaptic neurons (Supplementary Fig. 15b, c). However, this effect could be prevented by AMPA receptor antagonist NBQX but not by a cocktail of D1 and D2 receptor antagonists (Supplementary Fig. 15). Technically, electrophysiological recordings have limitations in capturing all postsynaptic neurons. A recent fiber photometry study indicated the co-release of glutamine and dopamine in the NAc shell by VTA glutamatergic neurons[49]. Therefore, so far, we cannot exclude the possibility of dopamine co-release and its functional role in regulating STA behavior, but we believe that glutamatergic neurotransmission plays a predominant role in our circuitry studies. Our further analysis showed that both D1 receptor- and D2 receptor-positive neurons in the NAc shell and LHb responded to optogenetic stimulation (Supplementary Fig. 15). Future studies focusing on cell-type-specific neuronal activity and function studies will presumably provide a more comprehensive understanding of the

precise role of postsynaptic neuronal subtypes in mediating STA inter-individual differences.

Previous studies have predominantly focused on the mechanisms underlying pain susceptibility and related pathology, whereas less attention has been paid to pain resilience[1]. Recent pioneering pain resilience investigations, particularly in patients with IEM, have revealed varying pain experience among individuals. Notably, some IEM patients reported diminished pain, while others experienced severe pain. In these patients, KCNQ2 subtype of K[+] channels was identified in the dorsal root ganglia and emerged as a peripheral mechanism contributing to pain resilience, suggesting KCNQ channels as potential targets for promoting pain resilience and achieve therapeutic efficacy of analgesics[1,11–15]. Consistent with these findings, our research and others consistently reported the analgesic effectiveness of KCNQ channel opener, retigabine, in animal models of chronic inflammatory and neuropathic pain[13,17,18]. These lines of evidence may support that promoting resilience mechanisms represents a fundamentally different strategy for pain management, which would be unable to be achieved by solely focusing on pain susceptibility and pathology.

In addition to the different pain responses, BY mice might also develop other pain-related emotional behaviors during the social contact procedure, for example, anxiety- and depressive-like behaviors. Our c-Fos protein expression data demonstrated that some brain regions displayed similar changes in BY-S and BY-R mice. For instance, all BY mice exhibited increased c-Fos protein expression in their ACC, paraventricular thalamic nucleus, and LC. These data indicated possible common behavioral adaptations in all BY mice following the STA paradigm. Moreover, female mice displayed similar susceptible and resilient behavioral adaptations following the STA paradigm, indicating similar neurobiology underlying the inter-individual differences in female pain responses. The exact cellular and circuitry mechanisms under the inter-individual differences in female pain responses need further investigation in future independent studies. And the functional role of the VTA glutamatergic neurons in mediating resilience to other empathic behaviors or other stressful stimuli remains unknown and needs further studies.

In conclusion, this study establishes a different strategy for future research on the inter-individual difference in pain responses and identifies STA resilience-specific cellular and neural circuits in the brain, which provide beneficial information for the promotion of resilience to STA and the development of conceptually innovative analgesics.

## Methods

### Animal and housing

In this study, adult (age 8–14 weeks) male and female C57BL/6J, *Vglut2-IRES-Cre* (*Slc17a6^{tm2(cre)Lowl/J}*, stock no. 016963), and *Vgat-IRES-Cre* (*Slc32a1^{tm2(cre)Lowl/J}*, stock no. 016962) mice (purchased from Jackson Laboratories and bred onto a C57BL/6J genetic background) were used (male, 1791 animals; female, 50 animals). Animals were housed under standard laboratory conditions (12 h light/12 h dark cycle, lights on from 08:00 to 20:00, temperature of $23 \pm 2\,°C$, and humidity of 55–60%) with ad libitum access to standard lab mouse pellet food and water. No food or water is available during modeling and testing (1–2 h). All testing procedures were conducted during the light cycle between 9 a.m. and 4 p.m. All the mice were housed in standard mouse cages (4 per cage in a colony) before experiments and were singly housed following the social transfer paradigm. Age-matched animals were randomly assigned into different experimental groups, and all experiments were performed in accordance with protocols approved by the Institutional Animal Care and Use Committee of Xuzhou Medical University (the protocol number: 202208S103). Efforts were made to minimize animal suffering and reduce the number of animals used. Based on the similar behavioral outcomes observed in males and

females, we did not include females in our brain region screening and functional studies.

### Viral reagents and stereotactic surgeries

All surgeries were conducted under aseptic conditions. Mice were deeply anesthetized with an intraperitoneal injection of 1% sodium pentobarbital (40 mg/kg, i.p.) and placed in a stereotaxic frame (RWD Life Technology Co., LTD, Shenzhen, China). The eyes of the mice were kept moist using ophthalmic ointment throughout the surgery. The skull plane was adjusted to ensure that the bregma and lambda were at a horizontal level. Small holes were drilled in the skull above the target brain region using a dental drill to lower a syringe needle (5 μL, 33-gauge, Hamilton, Reno, NV, USA) into the target site. Viral vectors were bilaterally injected into the VTA, NAc shell or LHb following the coordinates (relative to the Bregma) below: VTA: $ML = \pm 0.4$ mm, $AP = -3.30$ mm, $DV = -4.40$ mm, 200 nl + 200 nl; NAc shell: $ML = \pm 0.55$ mm, $AP = +1.60$ mm, $DV = -4.64$ mm, 200 nl + 200 nl; LHb: $ML = \pm 0.43$ mm, $AP = -1.30$ mm, $DV = -2.85$ mm, 120 nl + 120 nl. The tip of the injector was lowered an additional 0.1 mm below the intended injection site and then raised to the final coordinate before injection. The virus was infused at a rate of 0.1 μL/min. The injection needle was left in place for an additional 10 min after the end of infusion and then withdrawn slowly to avoid back-flow. Mice were allowed to recover for at least 3 weeks before the behavioral experiments. The efficacy of viral expression was confirmed through histological examination at the end of the experiments. The mice with off-target EGFP or mCherry locations were excluded from the analysis. The virus used in this study was purchased from BrainVTA Co., Ltd. (Wuhan, China).

To visualize the glutamatergic and GABAergic neurons, rAAV2/9-EF1a-DIO-EGFP-WPRE-hGH polyA ($5.55 \times 10^{12}$ viral genome/ml [vg/ml], 200 nl for each site, BrainVTA) was bilaterally injected into the VTA of *Vglut2-IRES-Cre* or *Vgat-IRES-Cre* mice. This surgery approach was also used to screen the downstream brain targets of VTA glutamatergic neurons in *Vglut2-IRES-Cre* mice.

To study the overlap of VTA→LHb and VTA→NAc shell projecting glutamatergic neurons, rAAV2/R-EF1a-DIO-mCherry-WPRE-hGH pA ($5.71 \times 10^{12}$ vg/ml, 120 nl for each site, BrainVTA) and rAAV2/R-EF1a-DIO-EGFP-WPRE-hGH polyA ($5.20 \times 10^{12}$ vg/ml, 200 nl for each site, BrainVTA) were bilaterally injected into the LHb or NAc shell of *Vglut2-IRES-Cre* mice, which could be taken up by the terminals at the site of injection and transported back to the soma to express the mCherry or EGFP.

For slice electrophysiology with optogenetic stimulation, rAAV2/9-D1-mCherry-WPRE-hGH pA ($2.00 \times 10^{12}$ vg/ml, 120 nl for each site, BrainVTA) and rAAV2/9-D2-mCherry-WPRE-hGH polyA ($2.04 \times 10^{12}$ vg/ml, 200 nl for each site, BrainVTA) were bilaterally injected into the LHb and NAc shell of *Vglut2-IRES-Cre* mice, and rAAV2/9-EF1a-DIO-ChR2-eYFP-WPRE-hGH polyA ($3.42 \times 10^{12}$ vg/ml, 200 nl for each site, BrainVTA) was bilaterally injected into the VTA of *Vglut2-IRES-Cre*.

For chemogenetic manipulation of the VTA Glu neurons, rAAV2/9-EF1a-DIO-hM3D(Gq)-mCherry-WPRE-hGH polyA ($5.27 \times 10^{12}$ vg/ml, 200 nl for each site, BrainVTA), rAAV2/9-EF1a-DIO-hM4D(Gi)-mCherry-WPRE-hGH polyA ($5.18 \times 10^{12}$ vg/ml, 200 nl for each site, BrainVTA) was injected into the VTA of *Vglut2-IRES-Cre* mice. The mice injected with rAAV2/9-EF1a-DIO-mCherry-WPRE-hGH polyA ($5.14 \times 10^{12}$ vg/ml, BrainVTA) virus at the same volume were used as controls. For chemogenetic manipulation of the VTA DA neurons, rAAV2/9-TH-hM3D(Gq)-mCherry-WPRE-hGH polyA ($5.45 \times 10^{12}$ vg/ml, 200 nl for each site, BrainVTA) and rAAV2/9-TH-hM4D(Gi)-mCherry-WPRE-hGH polyA ($5.15 \times 10^{12}$ vg/ml, 200 nl for each site, BrainVTA) were injected into the VTA of C57BL/6J mice. The mice injected with rAAV2/9-TH-mCherry-WPRE-hGH polyA ($5.59 \times 10^{12}$ vg/ml, BrainVTA) virus at the same volume were used as controls. For chemogenetic manipulation of the VTA→NAc shell or VTA→LHb glutamatergic projections, the FLP-dependent virus rAAV2/9-EF1a-fDIO-hM3D(Gq)-

EGFP-WPRE-hGH polyA ($5.16 \times 10^{12}$ vg/ml, 200 nl for each site, BrainVTA) or rAAV2/9-EF1a-fDIO-hM4D(Gi)-EGFP-WPRE-hGH polyA ($4.77 \times 10^{12}$ vg/ml, 200 nl for each site, BrainVTA) was delivered into the VTA, and rAAV2/R-EF1a-DIO-FLP-WPRE-hGH polyA ($5.93 \times 10^{12}$ vg/ml, BrainVTA) was delivered into the NAc shell or LHb of *Vglut2-IRES-Cre* mice. The mice injected with rAAV2/9-EF1a-fDIO-EGFP-WPRE-hGH polyA ($5.21 \times 10^{12}$ vg/ml, BrainVTA) virus into VTA at the same volume were used as controls.

## CFA inflammatory pain
CFA inflammatory pain was induced by injecting CFA[35] (10 ml, Beyotime P2036). Mice were lightly restrained and immediately received 10 µl of CFA unilaterally into the intra-plantar surface of the left hind paw to induce inflammatory pain. Behavioral tests were performed to evaluate the acute pain four hours following the injection.

## Social transfer of pain (single-time and repeated paradigms)
Following a recent protocol outlined by Smith et al.[35], mice were randomly paired and co-housed in group of 4 per cage for at least 7 days prior to each experiment. This housing procedure allows the familiarization of both bystander and CFA demonstrator mice before experimental manipulations. In single-time experiments, mice co-housed either in 4 per cage (comprising 2 BY mice and 2 CFA paired partners) or 4 per cage with 1 BY mouse and 3 CFA paired partners for chemogenetic repeated manipulation paradigms. Adequate air filtration is highly recommended, and the modeling and testing room should be thoroughly cleaned before testing to minimize potential confounding olfactory cues associated with pain, which can alter the behavior of untreated mice[37]. Thus, cages, bedding and other olfactory cues of experimental mice should never be left exposed in the test areas or animal colony. Gently return all mice to their home cages once they have become accustomed to the model room and testing equipment, then move them back to their original housing room.

All mice underwent a baseline von Frey test for mechanical allodynia on the day before social transfer. To examine the social transfer of pain, two mice (CFA) from each cage were harvested on the test day, lightly restrained, and immediately injected with 10 µl of CFA into the intra-plantar surface of the left hind paw, a procedure well-established for inducing enduring, arthritis-like pain[58,59]. After injection, each CFA mouse was placed alone into a clean bedding cage (without food or water) and subjected to a 1-h social contact with one of its paired cagemates, allowing the two mice to interact freely. BY mice were subjected to mechanical threshold testing at different time points according to the experimental paradigms. Control mice were placed as pairs in a clean bedding cage without other treatments. To avoid the influence of olfactory cues, modeling and subsequent mechanical testing of control mice must be performed separately (in separate rooms or in the same room at different times) from CFA and BY mice. After the initial behavioral assays, mice were singly housed for the following experiments. The secondary repeated paradigm of social transfer of pain was performed at a 7-day interval to allow for full recovery from the initial exposure.

## Separation of susceptible and resilient sub-populations
To test whether the mechanical allodynia test was able to separate the susceptible and resilient sub-populations in individual mice, an effective unsupervised learning method, *K*-means cluster analysis on the PWTs results was performed.

Cluster analysis was conducted on the immediate PWTs (in grams) and the ratio of the PWTs (PWTs ratios = immediate PWTs after modeling/baseline PWTs before modeling * 100, %) after modeling to generate two independent cutoff values: PWTs = 0.41 g and PWTs ratio = 75%. These cutoffs were used according to the experimental design, which had been carefully described in the main text.

## Von Frey test for mechanical allodynia
Responses to mechanical stimulation by von Frey hairs (von Frey hairs; Stoelting, Kiel, WI, USA) (0.008, 0.02, 0.07, 0.16, 0.4, 1, 2 and 6 g) were determined in the plantar surface of the left hind paw. Positive responses included licking, biting, shaking and sudden withdrawal of the hind paws; otherwise, negative. Mechanical thresholds were tested using the up–down method[60]. Stimuli were always presented consecutively, whether ascending or descending. Starting from 0.16 g, in the absence of a positive response to the initially selected hair, a stronger stimulus was presented; in the event of a positive response, the next weaker stimulus was chosen until the first positive and negative were crossed, and then measured 4 times in a row (more than 30 s between each stimulus). The resulting pattern of positive and negative responses was tabulated using the convention, $X$ = positive response; $O$ = negative response, and the response threshold was interpolated using the formula: PWTs $= 10^{(X_f + k\delta)}$, where $X_f$ = value (in log units) of the final von Frey hair used; $k$ = the coefficient of different stimuli for the pattern of positive/negative responses; and $\delta$ = mean difference (in log units) between stimuli (here, 0.410723). Mice were pre-acclimated to the testing environment for 2 days before baseline testing and then placed individually under inverted clear plexiglass boxes (length: 8 cm × width: 8 cm × height: 5.5 cm split into four quadrants) on an elevated metal mesh rack and allowed to habituate for 40 min before each testing[61–63]. All nociceptive sensitizations described here are of relatively short duration (1–10 min) and allow the mouse to withdraw its paw from the painful stimulus. All tests were performed in a blinded manner.

## Social interaction test
A two-stage social interaction test was performed in a squared arena (40 cm × 40 cm) with artificially defined interaction zone (14 cm × 26 cm) and corner zones (10 cm × 10 cm) as we previously reported[30]. In the first test (the target was an unfamiliar naive mouse), the experimental mouse was allowed to freely explore the arena with an empty wire mesh sleeve (10 cm × 6 cm) in the interaction zone. In the second test, the experimental mouse was reintroduced into the arena with an unfamiliar CFA mouse in the mesh sleeve. Adapted from the method previously reported, the social interaction ratio was calculated as time in the interaction zone with CFA target/time in the interaction zone with naive target × 100. Video-tracking software (ANY-maze, version 4.84, Stoelting Co., IL, USA) was used to record the time the experimental mouse spent in the interaction zone and corner zones.

## Targeted allolicking and allogrooming behavior assessment
Following a recently reported protocol[64], targeted allolicking was defined as visible licking by a bystander mouse directed toward another home cage mouse's CFA- or saline-injected paw. Allogrooming was defined as visible licking and/or mouth contact localized on another mouse's body trunk, shoulder region and head.

## Immunohistochemistry
Mice were deeply anesthetized with an intraperitoneal injection of 1% sodium pentobarbital (40 mg/kg, i.p.) and transcardially flushed with 40 ml of 0.01 M cold phosphate-buffered saline (PBS, pH = 7.4), followed by 20 ml of PBS containing 4% paraformaldehyde. Immediately after perfusion, the brains were carefully removed and post-fixed in 4% paraformaldehyde at 4 °C for another 24 h and then dehydrated in 20% sucrose-PBS solution for 24 h, followed by 30% sucrose for 24 h. Coronal sections (35 µm) were prepared using a cryostat microtome (Leica VT1000S, Germany). For immunohistochemistry, these slices were rinsed in PBS three times for 5 min, followed by blocking of non-specific reactions with 1% bovine serum albumin in PBS containing 0.4% Triton X-100 (TBS) for 45 min at room temperature. Then, these slices were incubated overnight with appropriate primary antibodies diluted in TBS at 4 °C in a shaker. The primary antibodies included

rabbit anti-c-Fos (1:1000, Cell Signaling Technology, 2250) and mouse anti-TH (1:1000, Sigma, MAB318). Incubated brain slices were washed three times in TBS for 10 min and incubated for 2 h with the corresponding fluorophore-conjugated secondary antibody in TBS and subsequently washed twice in TBS for 10 min each and once in PBS for 10 min at room temperature. The secondary antibodies included anti-mouse Alexa 488 (1:500, Thermo Fisher Scientific, A21202) and anti-rabbit Alexa 594 (1:500, Thermo Fisher Scientific, A21207). Finally, all slices were mounted on glass slides and captured using a confocal laser-scanning microscope (LSM880, Carl Zeiss, Germany; FV1000, Olympus, Japan). For analysis of the counts of Fos+ cells, the target brain regions were encompassed by manually drawing boundaries according to the brain atlas (Franklin and Paxinos, *The Mouse Brain in Stereotaxic Coordinates*, 4th Edition, 2013), and cells were manually counted from at least 2–3 slices/mice.

### Slice electrophysiology with and without optogenetic stimulation

For neuronal firing activity recordings, 1.5 h following the STA paradigm and von Frey behavioral tests, mice were deeply anesthetized with 1% pentobarbital (40 mg/kg, i.p.) and perfused with oxygenated (95% $O_2$ and 5% $CO_2$) artificial cerebrospinal fluid (aCSF) containing 128 mM NaCl, 3 mM KCl, 1.25 mM $NaH_2PO_4$, 10 mM D-glucose, 25 mM $NaHCO_3$, 2 mM $CaCl_2$ and 2 mM $MgSO_4$. Acute brain slices were prepared using a microslicer in sucrose-enriched aCSF (3 mM KCl, 1.25 mM $NaH_2PO_4$, 10 mM D-glucose, 254 mM sucrose, 25 mM $NaHCO_3$, 2 mM $CaCl_2$ and 2 mM $MgSO_4$), incubated in aCSF for 30 min at 33 °C and then maintained at room temperature for a stable period of 1 h. Slices with the VTA were then transferred to a recording chamber with oxygenated aCSF at room temperature, flowing at a constant rate (2.5 ml/min). Glutamatergic neurons were identified based on mCherry expression and location under an Olympus BX61 microscope. Glass recording pipettes (6–8 MΩ) were filled with an internal solution composed of 130 mM K-methanesulfate, 10 mM KCl, 10 mM HEPES, 0.4 mM EGTA, 3 mM MgATP, 0.5 mM $Na_3$GTP and 7.5 mM phosphocreatine $(Na_2)$. Cell-attached recordings were performed to record the firing activity of VTA glutamatergic neurons.

For the recordings of optogenetic stimulation-induced excitatory postsynaptic currents (oEPSCs), a whole-cell patch clamp was performed in mCherry-labeled postsynaptic D1 receptor- or D2 receptor-positive neurons in the NAc shell or LHb. Once the postsynaptic neurons responded to optogenetic stimulation (473 nm, 10 ms in width, 3–5 mW), oEPSCs were recorded with control ACSF, D1 and D2 receptor antagonists (diluted in ACSF, D1 receptor antagonist SCH-23390 in 1 mM, MedChemExpress, HY-19545A; D2/3 receptor antagonist raclopride in 1 mM, MedChemExpress, HY-103414) or AMPA receptor antagonist (NBQX diluted in ACSF, 10 µM, Sigma, N171). Light pulses were repetitively applied every 60 s for up to five times, and oEPSCs amplitude were averaged over these stimuli. Data acquisition was collected using a Multiclamp 700B, and a Digidata 1440A digitizer and analyzed with Clampfit10.4 (Molecular Devices).

### Statistics

The data obtained from the mice with missed injections were excluded from further analysis by experimenters blinded to the experimental conditions. The *Shapiro–Wilk test* was used to check the normality of data. Data that did not pass the normality test were compared using a nonparametric test. Normally distributed data with equal variance were compared using an unpaired two-sided *t*-test. For multiple comparisons, one- or two-way analysis of variance (ANOVA), followed by Tukey's post hoc test for multiple comparisons test to determine statistical significance. Nonparametric Mann–Whitney test was applied to the data between two groups, the Kruskal–Wallis test was used for comparison between multiple groups, and the two-sided Friedman test was conducted for statistical evaluation of data between multiple

correlation groups with Dunn's multiple comparisons test if quantitative data were not normally distributed. The two-sided Fisher's exact test and chi-square test were employed for the percentage comparison. All data in this study are presented as mean ± s.e.m. The significance levels are indicated as *$P < 0.05$, **$P < 0.01$, ***$P < 0.001$, ****$P < 0.0001$. GraphPad Prism 8 (Graph Pad Software, Inc.), SPSS 16.0 (IBM, Inc.), Clampfit 10.4 (Sunnyvale, Inc.), and R 4.2.2 (Lucent, Inc.) were used for statistical analyses and graphing.

### Reporting summary

Further information on research design is available in the Nature Portfolio Reporting Summary linked to this article.

## Data availability

Source data are provided with this paper.

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

## Acknowledgements

This study is supported by the National Key Research and Development Program of China Brain Science and Brain-Like Intelligence Technology 2021ZD0203100 (J.-L.C.), National Key Research and Development Program of China Brain Science and Brain-Like Intelligence Technology 2021ZD0202900 and 2021ZD0202902 (M.-H.H.), National Natural Science Foundation of China 31970937 and 82271255 (H.Z.), 82301413 (L.S.), 82171227 (H.L.), 82101315 (M.C.), Jiangsu Province Innovative and Entrepreneurial Team Program (H.Z.), Jiangsu Province Key R&D Program Social Development Project (BE2023690), the Natural Science Foundation of Zhejiang Province LY22H090019 (H.L.), the Priority Academic Program Development of Jiangsu Higher Education Institutions 21KJB320001 (M.C.), China Postdoctoral Science Foundation 2023M732973 (L.S.) and 2022M722676 (M.C.), Postgraduate Research and Practice Innovation Program of Jiangsu Province KYCX22_2918 (Y.H.), KYCX23_2952 (L.A.), KYCX22_2932 (X.Z.), Research Fund for International Senior Scientists T2250710685 (M.-H.H.), Shenzhen Natural Science Foundation JCYJ20220818101600001 (M.-H.H.), Shenzhen Key Laboratory of Precision Diagnosis and Treatment of Depression ZDSYS20220606100606014 (M.-H.H.), Shenzhen Medical Research Fund SMRF B2303012 (M.-H.H.), Science and Technology Research and Development Foundation of Shenzhen (High-level Talent Innovation and Entrepreneurship Plan of Shenzhen Team Funding) KQTD20221101093608028 (M.-H.H.).

## Author contributions

Conceptualization: H.Z., Y.H.; methodology: H.Z., Y.H.; investigation: Y.H., L.A., L.S., Y.Z., D.C., S.S., R.J., Q.L., Q.B., X.P., X.Z., M.C., J.Y.; funding acquisition: H.Z., J.-L.C., M.-H.H., H.L., Y.H., L.A., X.Z.; project administration: H.Z., J.-L.C.; supervision: H.Z., J.-L.C., M.-H.H., H.L., A.H.; writing – original draft: H.Z., Y.H.; writing – review & editing: H.Z., Y.H., D.C., M.-H.H., J.-L.C.; revision: Y.H., J.D., L.S., L.A., S.S., J.-L.C., M.-H.H. and H.Z.; Y.H., L.A., L.S. and Y.Z. contributed equally to this work.

## Competing interests

The authors declare no competing interests.
