## [Peer Review File · Nature Communications]

Midbrain glutamatergic circuit mechanism of resilience to socially transferred allodynia in male miceREVIEWER COMMENTS

Reviewer #1 (Remarks to the Author):

Thanks for inviting me to review this interesting manuscript entitled "Midbrain glutamatergic circuit mechanism of resilience to pain" (NCOMMS-23-36972-T). This manuscript touches on a midbrain glutamatergic circuit for resilience to pain, which is an important continuum of the lab's previous studies concerning midbrain and limbic mechanisms of pain. With an exquisite experimental design, solid evidence was procured via a combination of chemogenetics, FOS staining, and behavioural tests. The scientific merits of this manuscript are exemplified in following three aspects. First, the authors established a novel behavioral paradigm that addressed an important question, inter-individual difference in pain response; Second, a specific cluster of midbrain neurons is identified to mediate this phenomenon; Third, related downstream circuitry for the development and maintenance of this phenomenon is further identified. The manuscript has a good logic, is well-written, and thus meets the criteria of publication in this honored journal. However, a minor revision should be performed before acceptance.

Several important issues should be critically considered:

1. The rationale of the establishment of the novel behavioral model should be clarified. The authors utilized a STA paradigm, and it reflects the ability of animals to develop empathetic pain. In my opinion, it may better fall into the category of psychology concerning one's psychological trait of emotional indifference. What about traditional nerve injury or inflammatory pain model? In our experience, some rodents recovered quickly from initial hyperalgesia induced by these treatments.
2. Pain resilience may not only be exemplified in the sensory aspect pain. The resilience to pain-related emotional disorders should not be overlooked.
3. The authors examined the PWT of the observers after STA paradigm. In my opinion, using pain threshold test to reflect empathetic pain sometimes carries subjective errors. Have the authors noticed interactive behaviors exerted by the observers to the sufferers, such as allo-grooming, licking, and caress in this process?
4. Current neurobiological mechanisms underlying pain resilience, from genetic or imaging view of points, should be elaborated in detail in the part of Introduction. This will allow a better understanding of study background for readers interested in this topic.
5. This study highlight the role of VTA glutamatergic neurons (but not dopaminergic neurons) in empathetic pain. Would the authors compare these two clusters of cells in the VTA, from the aspects of neurochemical anatomy, inputs and outputs, neurobiological functions in the Discussion? These knowledge may help readers better comprehend the discrete roles of these neurons in pain modulation.
6. The authors identified VTA(glu)-NAc and VTA(glu)-LHb projections involved in pain resilience, however, the postsynaptic neurons of these pathways remain unknown. Whether they share the same neurochemical features with their dopaminergic counterparts?

Reviewer #3 (Remarks to the Author):

'Midbrain glutamatergic circuit mechanism of resilience to pain' by Han et al. utilized a socially transferred allodynia (STA) paradigm and divided STA-treated mice into pain susceptible and resilient subgroups. By mapping the different fos expression and utilizing chemogenetic manipulation, they found that VTA-glu neurons are important for the regulation of pain resilience. Finally, VTA-NAc and VTA-LHb pathways are confirmed to respectively regulate the development and maintenance of pain resilience.

The subject of pain resilience and susceptibility is an interesting and important topic in the field of pain research, and it has been observed in clinical cases. However, given that pain is a basic innate response in animals, it is challenging to establish a pain model where some animals experience pain

while others do not. In this experiment, by establishing an observational pain model, the authors have successfully solved this problem. This is a delicate design, and it is obvious innovative. In total, the experiments were well performed. The experimental results are reliable. The manuscript is well written in general. However, I have several concerns need to be addressed. I think those should be helpful to improve the quality of the work.

Major:

1. Inhibition of the VTA-glu neurons increases the observational pain susceptibility. However, it also induced CFA-induced allodynia. Therefore, it's naturally to ask whether inhibition of glutamate neurons lead to body pain sensitivity or weaken the resistance to pain? In fact, if the activation of glutamate produces an analgesic-like effect, then the results in the study are consistent with this observation. I think confirming the VTA-glu neurons also regulate the resilience for other stress-stimulation may be helpful. Whether this is true or not, the meaning and topic of the work should be slightly modified. They should explain the meaning of choosing the observational pain model in the Introduction and thoroughly discuss this point.
2. Fos mapping data show that the Fos expression in the VTA is increased due to the enhanced Glu activities in the resilience group. However, why is the Fos expression in the NAc and LHb (the two main downstream targets of VTA-Glu neurons) decreased?
3. In this study, the authors proved that activation of VTA-NAc glu neurons induced analgesic effect. However, a previous study from the same group finds an opposite results, showing that inhibition of the VTA-NAc glu projections relieved neuropathic pain and inflammatory pain (Front Mol Neurosci. 2022; 15: 1083671).

Minor:

1. Page 3, line 5, '~70% of BY mice displayed decreased PWTs, and the rest of the ~30% animals exhibited comparable PWTs with the control mice (Fig. 1c)'. Fig. 1c should be changed to Figs. 1c and e, since the percentage (~70%) is the valuation from the two figures.
2. What do the arrows indicate to in Figs. 2f, h and j? Please explain that in the figure legend. I also noticed that many arrows are not indicating any neurons. The sharpness of Fig. 2d is not satisfied too.
3. In Fig. 5i, it's indicated that chemogenetic inhibition of the VTA-NAc shell glutamatergic projection 'induced more BY-S individuals'. However, from the figure, it should be all mice are BY-S mice.

Reviewer #4 (Remarks to the Author):

The manuscript by Han et al. investigated inter-individual differences in pain response, especially pain resilience, by establishing two independent cutoffs to separate the susceptible and resilient sub-populations in a social transfer pain model. The behavioral tests provided solid evidence to support that the susceptible and resilient behavioral phenotypes are stable. Further functional studies revealed VTA glutamatergic neurons as a cellular target for pain resilience and identified the VTA-NAc shell and the VTA-LHb projections as two independent circuitry mechanisms for the development and maintenance of pain resilience, respectively. The manuscript defined and separated the susceptible and resilient individuals in the social transfer model of pain, thus providing an animal model tool to investigate inter-individual differences in pain response. The mechanism study illuminates a comprehensive picture describing the important role of mesolimbic reward system in pain resilience. Overall, this study provides a conceptual advance for pain research field, the experiments are well designed and data are well presented.

I have a few comments for the authors to further improve the manuscript.

1. The C57BL/6J bystander (BY) mice displayed different pain behaviors after a brief social interaction with CFA age mates. Is the inter-individual difference in pain responses from the difference in social interaction behaviors? Would the susceptible mice spend more time interacting with CFA tool mice?
2. c-Fos expression was used to examine VTA glutamatergic neuron activity in control, BY-S, and BY-R mice. Their results suggested an elevated c-Fos expression only in the VTA glutamatergic neurons of

BY-S mice. Other cell-type-specific approaches, such as electrophysiology, will provide more direct evidence to support their findings further.

3. The authors also touched on the behavioral outcomes in female mice exposed to this STA paradigm. Can female mice share exact cellular and circuitry mechanisms for their inter-individual difference in pain behaviors? It would be helpful to provide direct evidence or more discussions.

4. VTA glutamatergic neurons were activated in BY-R mice, and chemogenetic activation of the VTA-NAc shell glutamatergic projection promotes resilience. However, the c-Fos data only demonstrated an increase in the NAc shell of BY-S group but not the BY-R group. Please interpret this point.

5. Different neuronal responses were observed in different brain regions in BY mice. Some brain regions exhibited increased c-Fos protein in both BY subpopulations, and some displayed changed c-Fos protein only in the BY-R or BY-S mice. How will the author interpret these c-Fos expression patterns? Should other behavioral adaptations happen in both BY-R and BY-S mice?

Point-to-point response to reviewer's comments

We thank the reviewers for their insightful and constructive comments on the manuscript. We have now thoroughly addressed the reviewers' concerns and made substantial revisions accordingly.

Below we detailed the changes and point-by-point responses to the comments.

REVIEWER COMMENTS

Reviewer #1 (Remarks to the Author):

Thanks for inviting me to review this interesting manuscript entitled "Midbrain glutamatergic circuit mechanism of resilience to pain" (NCOMMS-23-36972-T). This manuscript touches on a midbrain glutamatergic circuit for resilience to pain, which is an important continuum of the lab's previous studies concerning midbrain and limbic mechanisms of pain. With an exquisite experimental design, solid evidence was procured via a combination of chemogenetics, FOS staining, and behavioural tests. The scientific merits of this manuscript are exemplified in following three aspects. First, the authors established a novel behavioral paradigm that addressed an important question, inter-individual difference in pain response; Second, a specific cluster of midbrain neurons is identified to mediate this phenomenon; Third, related downstream circuitry for the development and maintenance of this phenomenon is further identified. The manuscript has a good logic, is well-written, and thus meets the criteria of publication in this honored journal. However, a minor revision should be performed before acceptance.

We greatly appreciate your very positive comments and constructive advice on our manuscript. Please see below our responses to your comments and advice.

Several important issues should be critically considered:

1. The rationale of the establishment of the novel behavioral model should be clarified. The authors utilized a STA paradigm, and it reflects the ability of animals to develop empathetic pain. In my opinion, it may better fall into the category of psychology concerning one's psychological trait of emotional indifference. What about traditional nerve injury or inflammatory pain model? In our experience, some rodents recovered quickly from initial hyperalgesia induced by these treatments.

Thank you for your very thoughtful comments.

As we described in the original version, the present study was designed to look at the neuronal and circuitry mechanisms underlying the inter-individual differences in pain responses, especially the resilience to pain perception. To do this, the first step is recapitulating different behavioral phenotypes in lab animals. Because current pain models were established by inducing severe tissue damage (peripheral nerve ligation or skin incision) or local inflammation, it is hard to recapitulate susceptible versus resilient

phenotypes with these models (1-4). Recent animal studies demonstrate that the state of hyperalgesia could be socially transferred from one individual to another through a brief empathetic social contact, a pain model referred as socially transferred pain (5-9). Because social interaction with a CFA inflammatory conspecific is a relatively mild stress that could induce pain-like behaviors and the evident inter-individual difference in empathy and social behaviors (10-12) , we try to use the social transfer model of allodynia to model different pain profiles. And our findings strongly support that it is a workable animal model to study the development and maintenance of pain susceptibility and resilience.

To address the concern, the following information was added to the revised Introduction following the first paragraph:

“However, how the resilient brain copes with pain experience is not well known, partially due to the lack of experimental paradigms that permit stable replication of inter-individual differences in pain responses in laboratory animals. The establishment with severe tissue damage or inflammation makes it an obstacle to recapitulate susceptible versus resilient phenotypes in current animal pain models, such as the CFA-induced persistent inflammatory pain and the sciatic nerve ligation-induced neuropathic pain (1-4). Recent animal studies demonstrate that the state of hyperalgesia could be socially transferred from one individual to another through a brief empathetic social contact, a pain model referred to as socially transferred pain (5-9). Given the evident inter-individual difference in empathy processing and social behaviors (10-12) , this model provides an opportunity to model different pain profiles.”

We agree that some animals possibly recovered faster from hyperalgesia induced by traditional pain models than others. This is also an important scientific question that deserves systematic investigations, which would provide useful information for future therapeutic strategy development by promoting recovery mechanisms. We did not employ traditional pain models based on the scientific questions that need to be addressed: the development and maintenance of inter-individual differences in pain response.

2. Pain resilience may not only be exemplified in the sensory aspect pain. The resilience to pain-related emotional disorders should not be overlooked.

This is an essential question that needs to be studied. In addition to the inter-individual differences in pain perception, we believe that the BY mice also develop other behavioral outcomes following the one-hour brief social contact with their CFA cage mates. Our c-Fos protein expression data demonstrated that some brain regions displayed similar changes in BY-S and BY-R mice, indicating identical behavioral adaptations in these subpopulations. These data suggest that BY mice might develop common emotional disorders during social contact. Here, we performed elevated plus maze and open field tests, two measurements for anxiety behaviors, and observed decreased open arm time in the elevated plus maze test. However, no significant behavioral adaptations were observed in the open field test. These preliminary data suggest anxiety-like behaviors in

both BY-S and BY-R mice. These data were under evaluation in another journal, so we include it below in our rebuttal letter here, but not in our revised manuscript.

And in our revised manuscript, we added more discussions accordingly:

“In addition to the different pain responses, BY mice might also develop other pain-related emotional behaviors during the social contact procedure, for example, anxiety- and depressive-like behaviors. Our c-Fos protein expression data demonstrated that some brain regions displayed similar changes in BY-S and BY-R mice, indicating identical behavioral adaptations in these subpopulations. However, whether these behaviors developed in specific subgroups or all the BY mice is unknown.”

A Socially Transferred Allodynia(STA)

Anxiety-like behavior in both BY-S and BY-R mice. (A) Open field test and elevated plus maze test experimental Schematic. (B) Time spent in the open arms in the elevated plus maze test (Control, n =13 mice; BY-S, n = 10 mice; BY-R, n = 3 mice). (C) Open arm ratios (open arm time/ open arm time + closed arm time *100) (Control, n = 13 mice; BY-S, n = 10 mice; BY-R, n = 3 mice). (D) Distance in the open arms (Control, n =13 mice; BY-R, n = 10 mice; BY-S, n = 3 mice). (E) Center zone time in the open field test (Control, n =12 mice; BY-S, n = 9 mice; BY-R, n = 4 mice). (F) Periphery zone time (Control, n =12 mice; BY-S, n = 9 mice; BY-R, n = 4 mice). The data are presented as the *means ± SEM*. **P* < 0.05, ***P* < 0.01, ns: no significance. Data analyzed by (B, C, D, E and F) *One-way ANOVA with Tukey's multiple comparisons test*.

3. The authors examined the PWT of the observers after STA paradigm. In my opinion, using pain threshold test to reflect empathetic pain sometimes carries subjective errors. Have the authors noticed interactive behaviors exerted by the observers to the sufferers, such as allo-grooming, licking, and caress in this process?

Thank you for the comments.

We agree that evaluating pain responses with a threshold test will invite subjective errors. In the past three decades, paw withdrawal threshold, measured by the von Frey test, has been accepted and intensively used to evaluate mechanical allodynia in preclinical and clinical studies. And this pain threshold test was very well performed in our previous studies over the past decade. The evidence above suggests the PWTs measured with von Frey test are reliable for mechanical allodynia in our study.

And following your suggestion, we have performed social interaction test in the control, BY-S and BY-R mice. This social interaction test involved 2.5 minutes of environmental exploration in an open arena with a male naive mouse and another 2.5 minutes of social interaction with a target male C57BL/6J mouse with CFA inflammatory pain. And a group of CFA mice were also subjected to the interaction test. Behavioral tests demonstrated no significant differences in the social interaction time and locomotion among different groups (Supplementary Fig. 3).

To further examine the interactive behaviors between the BY mice and CFA mice during the STA paradigm, we analyzed the general allogrooming and targeted allolicking behaviors (injured paw-licking by the BY mice) during the 1 hour social contact. However, no significant difference between the BY-S and BY-R mice in the allogrooming and targeted allolicking behaviors was observed (Supplementary Fig. 4). Further detailed analysis performed every 15 min did not show significant differences between the two sub-groups in the general allogrooming and licking behaviors.

These data demonstrate that BY-S and BY-R mice display similar behaviors when exposed to social interaction with CFA inflammatory pain mice during and after the STA paradigm.

In the revised manuscript and the supplemental file, we provided the following information to address this point in the last paragraph of Result 1.

“Moreover, our behavioral tests during and after the STA paradigm suggested that both BY-S and BY-R mice exhibited similar social interaction, allogrooming and targeted allolicking behaviors (Supplementary Fig. 3-4), indicating the inter-individual difference in pain responses between the two subpopulations was not derived from the difference in social interactive behaviors.”

Supplementary Fig. 3 | BY-S and By-R mice displayed similar social interaction with CFA mice. **a**, Experimental timeline for modeling and social interaction (SI) test. **b**, Social interaction ratios (Control, $n = 18$ mice; BY-S, $n = 13$ mice; BY-R, $n = 7$ mice; CFA, $n = 20$ mice). **c**, Interaction zone time (Control, $n = 18$ mice; BY-S, $n = 13$ mice; BY-R, $n = 7$ mice; CFA, $n = 20$ mice). **d**, Total distance traveled (Control, $n = 18$ mice; BY, $n = 20$ mice; CFA, $n = 20$ mice). **e**, Total distance traveled (Control, $n = 18$ mice; BY-S, $n = 13$ mice; BY-R, $n = 7$ mice; CFA, $n = 20$ mice). The data are presented as the mean \pm s.e.m. Ns: no significance. Data analyzed by (b) Kruskal-Wallis test with Dunn's multiple comparisons

test; or (c, d, and e) two-way RM ANOVA with Tukey's multiple comparisons test. Details of the statistical analyses are presented in Supplementary Table 5.

Supplementary Fig. 4 | General allogrooming and targeted allolicking during the STA paradigm. **a**, Experimental timeline and modeling protocol for examining the behaviors of bystander mice during the STA. **b-e**, Total duration (**b, c**) and bouts number (**d, e**) of allogrooming and targeted allolicking behaviors during 1h STA (Control, $n = 6$ mice; BY-S, $n = 8$ mice; BY-R, $n = 6$ mice). **f-i**, Total duration (**f, g**) and bouts number (**h, i**) of allogrooming and targeted allolicking behaviors during the 0-15 minute period of STA (Control, $n = 6$ mice; BY-S, $n = 8$ mice; BY-R, $n = 6$ mice). **j-m**, Total duration (**j, k**) and bout number (**l, m**) of allogrooming and targeted allolicking behaviors during the 15-30 minute period of STA (Control, $n = 6$ mice; BY-S, $n = 8$ mice; BY-R, $n = 6$ mice). **n-q**, Total duration (**n, o**) and bout number (**p, q**) of allogrooming and targeted allolicking behaviors during the 30-45 minute period of STA (Control, $n = 6$ mice; BY-S, $n = 8$ mice; BY-R, $n = 6$ mice). **r-u**, Total duration (**r, s**) and bout number (**t, u**) of allogrooming and targeted allolicking behaviors during the 45-60 minute period of STA (Control, $n = 6$ mice; BY-S, $n = 8$ mice; BY-R, $n = 6$ mice). **v-y**, Cumulative duration (**v, w**) and bout number (**x, y**) of allogrooming and targeted allolicking behaviors in BY-S, BY-R, and control mice measured every 15-minute (Control, $n = 6$ mice; BY-S, $n = 8$ mice; BY-R, $n = 6$ mice). The data are expressed as the mean \pm s.e.m. * $P < 0.05$, ** $P < 0.01$, *** $P < 0.001$, ns: no significance. Data analyzed by (**b-k, m-s, u**) Kruskal-Wallis test with Dunn's multiple comparisons test, or (**l, t**) One-way ANOVA with Tukey's multiple comparisons test. (**v-y**) Two-way RM ANOVA with Tukey's post hoc test; Details of the statistical analyses are presented in Supplementary Table 5.

4. Current neurobiological mechanisms underlying pain resilience, from genetic or imaging view of points, should be elaborated in detail in the part of Introduction. This will allow a better understanding of study background for readers interested in this topic.

Thank you for the very constructive comments.

Following your suggestion, we have revised the Introduction by adding more information from recent genetic, functional magnetic resonance imaging, and animal neural circuitry studies. The following information was added to the first paragraph of the revised Introduction.

"The interest and attention on pain resilience and stress resilience have been dramatically increasing in recent decades. For example, KCNQ2 encoding inhibitory potassium (K⁺) channels in the dorsal root ganglia and VTA has been identified as a target for pain resilience and social stress resilience (13-18), which leads to the potential use of KCNQ channel opener, retigabine (also called ezogabine), as potential analgesic and antidepressant (19-23). Functional magnetic resonance imaging studies demonstrated significant differences in grey matter volume and functional connectivity within the default mode network underlying subjective reports of more or less resilient response to pain states (24-26), indicating anatomical and functional adaptations in the resilient individuals. These resilience studies open a new avenue to develop conceptually innovative therapies for major depressive disorder and pain."

5. This study highlights the role of VTA glutamatergic neurons (but not dopaminergic neurons) in empathetic pain. Would the authors compare these two clusters of cells in the VTA, from the aspects of neurochemical anatomy, inputs and outputs, neurobiological functions in the Discussion? These knowledge may help readers better comprehend the discrete roles of these neurons in pain modulation.

Thanks for the very fundamental question.

We have added related information in the revised discussion:

“VTA is a core brain region with dense dopaminergic neurons (~60%) and also contains a relatively small number of glutamatergic neurons (~5%) (27, 28). While both cell types have been implicated in mediating motivation-related behaviors, such as reward and aversion (29-32), the dopaminergic neurons were reported more for their functional roles in mediating stress-related mental disorders (10, 33-36). The VTA glutamatergic and dopaminergic neurons exhibit heterogeneity in their inputs, outputs, and neurobiological functions (37). The striatal regions and the pallidum contribute a more substantial portion of inputs to VTA dopaminergic neurons, whereas glutamatergic neurons predominantly receive cortical inputs (38). In rodents, the major targets of VTA dopamine efferents are the MSNs in the NAc. Additionally, VTA dopaminergic neurons also project to the amygdala, cortex, hippocampus, ventral pallidum, periaqueductal grey, bed nucleus of the stria terminalis, olfactory tubercle, and locus coeruleus (37). VTA glutamatergic neurons project to the NAc, LHb, ventral pallidum, and amygdala (39, 40). These anatomical differences might account for the functional differences in mediating behaviors.”

And please refer to our responses to your next question.

6. The authors identified VTA(glu)-NAc and VTA(glu)-LHb projections involved in pain resilience, however, the postsynaptic neurons of these pathways remain unknown. Whether they share the same neurochemical features with their dopaminergic counterparts?

Thank you for the question.

In our revised version, we have added more information in the Discussion section as follows:

“Our retrograde tracing experiment in vGlut2-Cre mice found that VTA glutamatergic neurons projecting to the nucleus accumbens and lateral habenula also express the dopaminergic neuron marker tyrosine hydroxylase (Supplementary Fig. 14), suggesting that these neurons may simultaneously release dopamine at their axon terminals in the NAc and LHb. Furthermore, our electrophysiological studies found that when optogenetically activating the axon terminals in the NAc or LHb induced excitatory postsynaptic currents in their postsynaptic neurons (Supplementary Fig. 15). However, this effect could not be prevented by AMPA receptor antagonist NBQX, but not the cocktail of D1 and D2 receptor antagonists (Supplementary Fig. 15). Technically, electrophysiological recordings cannot record all postsynaptic neurons. A recent fiber photometry study indicated co-release of glutamine and dopamine in the NAc shell by VTA glutamatergic neurons (29). Further analysis found that both D1 receptor-positive and D2

receptor-positive neurons in the nucleus accumbens and lateral habenula responded to optogenetic stimulation (Supplementary Fig. 15). With these data, we cannot exclude the possibility of dopamine co-release and its functional role in regulating STA behavior, but we believe that glutamatergic neurotransmission plays a predominant role in our circuitry studies. Future cell-type-specific neuronal activity and function studies will provide a more comprehensive understanding of the exact role of postsynaptic neuronal subtypes in mediating STA inter-individual differences.”

In an independent ongoing study, our preliminary data showed that activating these two subtypes of MSNs generated antagonistic behavioral results: whereas D2 MSNs promote, D1 MSNs prevent the development of STA. This data was not included in the revised manuscript.

Supplementary Fig. 14 | Co-expression of TH in VTA projecting glutamatergic neurons. **a**, Schematic for retrograde AAV-DIO-mCherry injection into the NAc shell of *Vglut2-IRES-Cre* mice. **b** and **c**, Representative immunofluorescent images and quantitative data for co-expression of TH in VTA-NAc shell projecting glutamatergic neurons. **d**, Schematic for retrograde AAV-DIO-mCherry injection into the LHb of *Vglut2-IRES-Cre* mice. **e** and **f**, Representative immunofluorescent images and quantitative data for co-expression of TH in VTA-LHb projecting glutamatergic neurons. White arrows indicate VTA projecting glutamatergic neurons co-expressing TH.

I

Supplementary Fig. 15 | Postsynaptic characteristics of VTA glutamatergic neurons in NAc shell and LHb. **a**, Schematic for anterograde AAV-DIO-ChR2-eYFP injection into the VTA and AAV-D1-mCherry or AAV-D2-mCherry into the NAc shell or LHb of *Vglut2-IRES-Cre* mice. Representative immunofluorescent images of virus expression in VTA, NAc and LHb were shown. **b**, Representative responsive traces and quantitative data for optogenetic stimulation of VTA glutamatergic terminals in NAc shell neurons in the presence of ACSF, D1/D2 antagonist, and NBQX, respectively. **c**, Representative responsive traces and quantitative data for optogenetic stimulation of VTA glutamatergic terminals in LHb neurons in the presence of ACSF, D1/D2 antagonist, and NBQX, respectively. **d**, Summary data of responding postsynaptic cells during optogenetic stimulation of VTA glutamatergic terminals in NAc shell and LHb.

Activation of NAcSh D1 or D2-type MSNs promotes resilience or susceptibility to socially transferred allodynia, respectively. **a**, PWTs for different subgroups of BY mice across three repeats of social transfer of allodynia paradigm with or without chemogenetic activation of NAcSh D1-type MSNs (Control, $n = 10$ mice; BY-S, $n = 7$ mice; BY-R, $n = 2$ mice). **b**, Bar graphs showing the percentages of BY-S and BY-R mice across the three repeats of the paradigm. **c**, PWTs for different subgroups of BY mice across three repeats of social transfer of allodynia paradigm with or without chemogenetic activation of NAcSh D2-type MSNs (Control, $n = 10$ mice; BY-S, $n = 6$ mice; BY-R, $n = 3$ mice). **d**, Bar graphs showing the percentages of BY-S and BY-R mice across the three repeats of the paradigm. The data are presented as the mean \pm s.e.m. * $P < 0.05$, ** $P < 0.01$, **** $P < 0.0001$, ns: no significance. Data analyzed by (a, c) Two-way RM ANOVA with Tukey's multiple comparisons test and (b, d) Fisher's exact test.

Reviewer #3 (Remarks to the Author):

'Midbrain glutamatergic circuit mechanism of resilience to pain' by Han et al. utilized a socially transferred allodynia (STA) paradigm and divided STA-treated mice into pain susceptible and resilient subgroups. By mapping the different fos expression and utilizing chemogenetic manipulation, they found that VTA-glu neurons are important for the regulation of pain resilience. Finally, VTA-NAc and VTA-LHb pathways are confirmed to respectively regulate the development and maintenance of pain resilience.

The subject of pain resilience and susceptibility is an interesting and important topic in the field of pain research, and it has been observed in clinical cases. However, given that pain is a basic innate response in animals, it is challenging to establish a pain model where some animals experience pain while others do not. In this experiment, by establishing an observation pain model, the authors have successfully solved this problem. This is a delicate design, and it is obvious innovative.

In total, the experiments were well performed. The experimental results are reliable. The manuscript is well written in general. However, I have several concerns need to be addressed. I think those should be helpful to improve the quality of the work.

We greatly appreciate your very positive comments and constructive advice on our manuscript. Please see below our responses to your comments and advice.

Major:

1. Inhibition of the VTA-glu neurons increases the observational pain susceptibility. However, it also induced CFA-induced allodynia. Therefore, it's naturally to ask whether inhibition of glutamate neurons lead to body pain sensitivity or weaken the resistance to pain?

Thank you for the question.

In the original manuscript, we demonstrated that chemogenetic inhibition of the VTA glutamatergic neurons increased STA susceptibility in BY mice (Fig. 4d and e). Furthermore, a remarkable pro-allodynia effect was also observed in the established BY-R mice (Fig. 4f-g). However, this pro-allodynia effect was not observed in the control mice (Fig. 4f-g). These data suggest that VTA glutamatergic neuron inhibition-induced pain-promoting effects are context-dependent. Chemogenetic inhibition induced mechanical allodynia in mice exposed to the 15 min sub-threshold paradigm further confirmed this hypothesis (Fig. 4h and i). These data suggests that a suppressed VTA glutamatergic neuronal activity primes the animal to respond to environmental painful stimuli, such as exposure to the sub-threshold STA paradigm.

In the revised manuscript, we also added a concluding sentence in the related Results section as follows:

"These results suggest that inhibiting VTA glutamatergic neuron exerts a susceptibility- and allodynia-promoting effect in a context-dependent manner."

In fact, if the activation of glutamate produces an analgesic-like effect, then the results in the study are consistent with this observation. I think confirming the VTA-glu neurons also regulate the resilience for other stress-stimulation may be helpful.

This is an interesting and important question.

Resilience represents an active adaptation in the face of adversity and provides hope for the development of novel therapeutics that mimic the natural resilience mechanisms. In the present study, we demonstrated the functional role of the VTA glutamatergic neurons in mediating resilience to pain. Currently, it is unknown if VTA glutamatergic neurons play a role in mediating resilience to other stressful stimuli. We also completely agree that the generalization of this finding in other pathological processing, if possible, is an essential scientific question needing future systematic investigations.

To emphasize the importance of this question, we added more discussions after the second paragraph of the Discussion section as follows:

“However, the functional role of the VTA glutamatergic neurons in mediating resilience to other empathic behaviors or stressful stimuli remains unknown and need further studies.”

Whether this is true or not, the meaning and topic of the work should be slightly modified. They should explain the meaning of choosing the observational pain model in the Introduction and thoroughly discuss this point.

Thank you for your very constructive comments.

In the revised Introduction, we added the following information to explain why the empathic pain model was used in this study following the first paragraph.

“However, how the resilient brain copes with pain experience is not well known, partially due to the lack of experimental paradigms that permit stable replication of inter-individual differences in pain responses in laboratory animals. The establishment with severe tissue damage or inflammation makes it an obstacle to recapitulate susceptible versus resilient phenotypes in current animal pain models, such as the CFA-induced persistent inflammatory pain and the sciatic nerve ligation-induced neuropathic pain (1-4). Recent animal studies demonstrate that the state of hyperalgesia could be socially transferred from one individual to another through a brief empathetic social contact, a pain model referred as socially transferred pain (5-9). Given the evident inter-individual difference in empathy processing and social behaviors (10-12) , this model provides an opportunity to model different pain profiles.”

2. Fos mapping data show that the Fos expression in the VTA is increased due to the enhanced Glu activities in the resilience group. However, why is the Fos expression in the NAc and LHb (the two main downstream targets of VTA-Glu neurons) decreased?

Thanks for your question.

As described in our original manuscript, VTA glutamatergic neurons were only activated in BY-R mice; VTA-NAc shell and VTA-LHb glutamatergic projection were responsible for the development and maintenance of STA resilience, respectively. Logically, the NAc shell and the LHb will express more c-Fos protein in the BY-R mice compared to the BY-S group. However, the c-Fos data only demonstrated an increase in the NAc shell of BY-S group but not the BY-R group. The divergence between the deduction and our experimental data was highly due to the functionally different neuronal subtypes in the NAc shell. The NAc shell contains medium spiny neurons (MSNs) divided into two subpopulations based on their expression of D1 or D2 dopamine receptors (D1 and D2 neurons). These two populations produce different or antagonistic behavioral outputs, for example, in mediating pain perception, cocaine intake and social stress-related depressive-like behaviors (41-43). Based on the above evidence, we hypothesized that c-Fos protein expression data observed in the present study was a mixed pattern including the changes of c-Fos expression in both D1 and D2 MSNs. In our ongoing study, our preliminary data showed that activating these two subtypes of MSNs generated antagonistic behavioral results: whereas D2 MSNs promote, D1 MSNs prevent the development of STA.

Activation of NAcSh D1 or D2-type MSNs promotes resilience or susceptibility to socially transferred allodynia, respectively. **a**, PWTs for different subgroups of BY mice across three repeats of STA with or without chemogenetic activation of NAcSh D1-type MSNs (Control, $n = 10$ mice; BY-S, $n = 7$ mice; BY-R, $n = 2$ mice). **b**, Bar graphs showing

the percentages of BY-S and BY-R mice across the three repeats of STA. **c**, PWTs for different subgroups of BY mice across three repeats of STA with or without chemogenetic activation of NAcSh D2-type MSNs (Control, $n = 10$ mice; BY-S, $n = 6$ mice; BY-R, $n = 3$ mice). **d**, Bar graphs showing the percentages of BY-S and BY-R mice across the three repeats of STA. The data are presented as the mean \pm s.e.m. * $P < 0.05$, ** $P < 0.01$, **** $P < 0.0001$, ns: no significance. Data analyzed by (a, c) Two-way RM ANOVA with Tukey's multiple comparisons test and (b, d) Fisher's exact test.

The LHb contains densely distributed vesicular glutamate transporter 2- and 3- (VGluT2 and VGluT3)-expressing glutamatergic neurons and a small number of GAD-65/67-expressing GABAergic neurons (44-46). As expected and as we described in our original manuscript, the LHb expressed more c-Fos protein in the BY-R group. According to its increased expression pattern in the LHb of BY-R mice, we hypothesized that the increased c-Fos protein observed in our study was mainly located in the glutamatergic neurons. Based on the expression of either D1 or D2 dopamine receptors, LHb neurons could also be divided into D1 receptor-positive neurons and D2 receptor-positive neurons.

Future cell-type-specific neuronal activity and function studies will provide a more comprehensive understanding of the exact role of postsynaptic neuronal subtypes in the NAc shell and LHb in mediating STA inter-individual differences.

3. In this study, the authors proved that activation of VTA-NAc glu neurons induced analgesic effect. However, a previous study from the same group finds an opposite results, showing that inhibition of the VTA-NAc glu projections relieved neuropathic pain and inflammatory pain (Front Mol Neurosci. 2022; 15: 1083671).

Thank you for the question.

We have noticed this contradiction.

To further confirm this finding. The chemogenetic activation experiments in overall and projection-specific VTA glutamatergic neurons were repeated by different students/technicians, and consistent results indicated a robust analgesic effect on CFA inflammatory pain when activated overall but not those projection-specific glutamatergic neurons in the VTA, as shown in the following figures which were not included in the revised manuscript.

Chemogenetic activation of the VTA glutamatergic neurons alleviates mechanical allodynia in CFA mice. **a**, PWTs for CFA mice at baseline, before, and after CNO injection obtained by experimenter A ($n = 12$ mice per group). **b**, PWTs for CFA mice at baseline, before, and after CNO injection obtained by experimenter B ($n = 7$ mice per group). **c**, PWTs for CFA mice at baseline, before, and after CNO injection obtained by experimenter C ($n = 5$ mice per group). The data are presented as the mean \pm s.e.m. ** $P < 0.01$, *** $P < 0.001$, ns: no significance. Data analyzed by (a, b) Friedman test with Dunn's multiple comparisons test; or (c) one-way RM ANOVA with Tukey's multiple comparisons test. Details of the statistical analyses are presented in Supplementary Table 5.

Chemogenetic activation of the VTA \rightarrow NAc shell glutamatergic projection does not affect the mechanical allodynia of CFA mice. **a**, PWTs for CFA mice at baseline, before, and after CNO injection obtained by experimenter A ($n = 10$ mice per group). **b**, PWTs for CFA mice at baseline, before, and after CNO injection obtained by experimenter B ($n = 5$ mice per group). The data are presented as the mean \pm s.e.m. * $P < 0.05$, *** $P < 0.001$, ns: no significance. Data analyzed by (a, b) Friedman test with Dunn's multiple comparisons test. Details of the statistical analyses are presented in Supplementary Table 5.

Minor:

1. Page 3, line 5, '~70% of BY mice displayed decreased PWTs, and the rest of the ~30%

animals exhibited comparable PWTs with the control mice (Fig. 1c)'. Fig. 1c should be changed to Figs. 1c and e, since the percentage (~70%) is the valuation from the two figures.

The related parts have been corrected in the revised version.

2. What do the arrows indicate to in Figs. 2f, h and j? Please explain that in the figure legend. I also noticed that many arrows are not indicating any neurons. The sharpness of Fig. 2d is not satisfied too.

The arrows in Fig. 2 indicate c-Fos-expressing neurons.

Some arrows were removed from the figures to avoid misunderstandings.

The figure is replaced with a new high-resolution one.

3. In Fig. 5i, it's indicated that chemogenetic inhibition of the VTA-NAc shell glutamatergic projection 'induced more BY-S individuals'. However, from the figure, it should be all mice are BY-S mice.

Thanks for the suggestion. In the revised version, we have replaced the sentence with "...whereas pathway-specific inhibition prevented the development of resilience and made all mice susceptible to STA."

Reviewer #4 (Remarks to the Author):

The manuscript by Han et al. investigated inter-individual differences in pain response, especially pain resilience, by establishing two independent cutoffs to separate the susceptible and resilient sub-populations in a social transfer pain model. The behavioral tests provided solid evidence to support that the susceptible and resilient behavioral phenotypes are stable. Further functional studies revealed VTA glutamatergic neurons as a cellular target for pain resilience and identified the VTA-NAc shell and the VTA-LHb projections as two independent circuitry mechanisms for the development and maintenance of pain resilience, respectively. The manuscript defined and separated the susceptible and resilient individuals in the social transfer model of pain, thus providing an animal model tool to investigate inter-individual differences in pain response. The mechanism study illuminates a comprehensive picture describing the important role of mesolimbic reward system in pain resilience. Overall, this study provides a conceptual advance for pain research field, the experiments are well designed and data are well presented.

We appreciate your very positive comments and constructive advice on our manuscript. Following your comments and suggestions, we have deeply revised our manuscript.

I have a few comments for the authors to further improve the manuscript.

1. The C57BL/6J bystander (BY) mice displayed different pain behaviors after a brief social interaction with CFA age mates. Is the inter-individual difference in pain responses from the difference in social interaction behaviors? Would the susceptible mice spend more time interacting with CFA tool mice?

This is a fundamental question.

To study possible differences in social interaction process between the BY-S and BY-R sub-groups, all of the BY mice underwent a two phase social interaction test on the day after the STA paradigm. This social interaction test involved 2.5 minutes of environmental exploration in an open arena with an unfamiliar naive male mouse and another 2.5 minutes of social interaction with a target male C57BL/6J mouse with CFA inflammatory pain. And a group of CFA mice were also subjected to the interaction test. Behavioral tests demonstrated no significant differences in the social interaction time and locomotion among different groups (Supplementary Fig. 3).

Please also refer to our response to the third question of the first reviewer.

2. c-Fos expression was used to examine VTA glutamatergic neuron activity in control, BY-S, and BY-R mice. Their results suggested an elevated c-Fos expression only in the VTA glutamatergic neurons of BY-S mice. Other cell-type-specific approaches, such as electrophysiology, will provide more direct evidence to support their findings further.

Thanks for this constructive advice.

In the revised version, we added firing activity data of VTA glutamatergic neurons from the control, BY-S and BY-R mice. We first labeled VTA glutamatergic neurons by locally injecting Cre-inducible AAV with mCherry in vGlut2-Cre mice. Three weeks later, mice were subjected to in vitro electrophysiology 1.5 hours following the STA paradigm. As expected, BY-R mice displayed an increased firing activity in their VTA glutamatergic neurons when compared with that in the control and BY-S mice (Supplementary Fig. 6). This finding further supported the hyperactivity of VTA glutamatergic neurons in the BY-R mice.

In the revised manuscript, we have added the following information in the Result section: "To further confirm the hyperactivity of VTA glutamatergic neurons in BY-R mice, we labeled these neurons with Cre-inducible AAV carrying mCherry in *Vglut2-IRES-Cre* mice for slice electrophysiology. Cell-attached recordings demonstrated an increased firing frequency of VTA mCherry-positive neurons in BY-R mice compared to that in the control or BY-S groups (Supplementary Fig. 6)."

Supplementary Fig. 6 | Increased firing activity in VTA glutamatergic neurons of the BY-R mice. **a**, Schematic illustration depicting viral injection, experimental design, and representative image of mCherry-positive VTA vGlut2⁺ cells during patch clamp recording. **b**, 50%PWTs (** $p < 0.01$, $n = 3, 3, 3$ mice) measured after social contact with CFA mice for 1h. **c-d**, *In vitro* patch clamp sample traces (**c**) and firing rates (**d**) of VTA vGlut2⁺ neurons from Control, BY-S and BY-R mice. * $p < 0.05$, ** $p < 0.01$; $n = 11, 11, 12$ cells from 3 mice/group.

3. The authors also touched on the behavioral outcomes in female mice exposed to this

STA paradigm. Can female mice share exact cellular and circuitry mechanisms for their inter-individual difference in pain behaviors? It would be helpful to provide direct evidence or more discussions.

Thank you so much for the question. This is an important point.

In our original manuscript, we demonstrated that female C57BL/6J mice displayed similar susceptible and resilient phenotypes following the 1-hour social contact process with a similar proportion as observed in the male mice (Supplementary Fig. 2). Based on the similar behavioral outcomes in male and females, we did not include females in our brain region screening and functional studies. And we agree that the cellular and circuitry mechanisms underlying the inter-individual differences in females are also essential scientific questions need to be addressed in future independent studies. To avoid misunderstanding, following the sex and gender reporting policies and your comments, we made related changes in our title and abstract to clarify that the main findings of the current study are obtained from male mice.

The title of the manuscript was changed to "*Midbrain glutamatergic circuit mechanism of resilience to pain in male mice*"

Related changes were also made in the Abstract.

In our revised manuscript, more discussions are added as follows:

"Moreover, female mice displayed similar susceptible and resilient behavioral adaptations following the STA paradigm, indicating similar neurobiology underlying the inter-individual differences in pain responses in females. The exact cellular and circuitry mechanisms under the inter-individual differences in female pain responses need further investigations in future independent studies."

4. VTA glutamatergic neurons were activated in BY-R mice, and chemogenetic activation of the VTA-NAc shell glutamatergic projection promotes resilience. However, the c-Fos data only demonstrated an increase in the NAc shell of BY-S group but not the BY-R group. Please interpret this point.

Thanks for your question.

As you mentioned above, VTA glutamatergic neurons were only activated in BY-R mice, and chemogenetic activation of the VTA-NAc shell glutamatergic projection promotes resilience. Logically, the NAc shell will express more c-Fos protein in the BY-R mice compared to the BY-S group. However, the c-Fos data only demonstrated an increase in the NAc shell of BY-S group but not the BY-R group. The divergence between the deduction and our experimental data was highly due to the functionally different neuronal subtypes in the NAc shell. The NAc shell contains medium spiny neurons (MSNs) divided into two subpopulations based on their expression of D1 or D2 dopamine receptors (D1 and D2 MSNs). These two populations produce different or antagonistic behavioral outputs, for example, in mediating pain perception, cocaine intake and social stress-related depressive-like symptoms (41-43). Based on the above evidence, we

hypothesized that c-Fos protein expression data observed in the present study was a mixed pattern including the changes of c-Fos expression in both D1 and D2 MSNs. Future cell type-specific neuronal activity and function studies will provide a more comprehensive understanding of the exact role of NAc shell subtypes in mediating STA inter-individual differences. In our ongoing study, our preliminary data showed that activating these two subtypes of MSNs generated antagonistic behavioral results: whereas D2 MSNs promote, D1 MSNs prevent the development of STA susceptibility.

Activation of NAcSh D1 or D2-type MSNs promotes resilience or susceptibility to socially transferred allodynia, respectively. **a**, PWTs for different subgroups of BY mice across three repeats of STA with or without chemogenetic activation of NAcSh D1-type MSNs (Control, $n = 10$ mice; BY-S, $n = 7$ mice; BY-R, $n = 2$ mice). **b**, Bar graphs showing the percentages of BY-S and BY-R mice across the three repeats of STA. **c**, PWTs for different subgroups of BY mice across three repeats of STA with or without chemogenetic activation of NAcSh D2-type MSNs (Control, $n = 10$ mice; BY-S, $n = 6$ mice; BY-R, $n = 3$ mice). **d**, Bar graphs showing the percentages of BY-S and BY-R mice across the three repeats of STA. The data are presented as the mean \pm s.e.m. * $P < 0.05$, ** $P < 0.01$, **** $P < 0.0001$, ns: no significance. Data analyzed by (a, c) Two-way RM ANOVA with Tukey's multiple comparisons test and (b, d) Fisher's exact test.

Please also refer to our response to the second question of reviewer 3.

5. Different neuronal responses were observed in different brain regions in BY mice. Some brain regions exhibited increased c-Fos protein in both BY subpopulations, and some displayed changed c-Fos protein only in the BY-R or BY-S mice. How will the author interpret these c-Fos expression patterns? Should other behavioral adaptations happen in both BY-R and BY-S mice?

Thank you for the comments. This is an important point.

Our immunofluorescent experiment identified some brain regions with specific changes of c-Fos protein expression only in BY-S or BY-R mice. For example, we observed increased c-Fos protein expression only in the VTA of BY-R mice, and increased c-Fos protein expression only in the dorsal raphe nucleus, anterior cingulate cortex and amygdala of BY-S mice. These results underscore a potentially specific role for these brain regions in regulating certain behavioral outcomes. Interestingly, some brain regions displayed similar changes in both BY-S and BY-R mice, indicating universal behavioral adaptations in these two subpopulations. Indeed, in the elevated plus maze test, a widely-used paradigm for measuring anxiety-like behaviors, we observed a significant decrease in the open arm time in both BY-S and By-R mice, indicating a possible anxiety-like behavior in the BY mice. However, both BY-S and BY-R mice showed no remarkable changes in the open field test, another behavioral test for anxiety measurement. And these data were under evaluation in another journal, so we include it in our rebuttal letter as follows, but not in our revised manuscript.

And in our revised manuscript, we added more discussions as follows:

“In addition to the different pain responses, BY mice might also develop other pain-related emotional behaviors during the social contact procedure, for example, anxiety- and depressive-like behaviors. Our c-Fos protein expression data demonstrated that some brain regions displayed similar changes in both BY-S and By-R mice. For example, all of the BY mice exhibited increased c-Fos protein expression in their anterior cingulate cortex, paraventricular thalamic nucleus, and locus coeruleus et.al. This data indicated possible common behavioral adaptations in all of the BY mice following the STA paradigm.”

A Socially Transferred Allodynia(STA)

Anxiety-like behaviors in both BY-S and BY-R mice. (A) Open field and elevated plus maze tests experimental Schematic. (B) Time spent in the open arms in the elevated plus maze test (Control, n = 13 mice; BY-S, n = 10 mice; BY-R, n = 3 mice). (C) Open arm ratios (open arm time/ open arm time + closed arm time *100) (Control, n = 13 mice; BY-S, n = 10 mice; BY-R, n = 3 mice). (D) Distance in the open arms (Control, n = 13 mice; BY-R, n = 10 mice; BY-S, n = 3 mice). (E) Center zone time in the open field test (Control, n = 12 mice; BY-S, n = 9 mice; BY-R, n = 4 mice). (F) Periphery zone time (Control, n = 12 mice; BY-S, n = 9 mice; BY-R, n = 4 mice). The data are presented as the *means ± SEM*. **P* < 0.05, ***P* < 0.01, ns: no significance. Data analyzed by (B, C, D, E and F) *One-way ANOVA with Tukey's multiple comparisons test*.

We thank the reviewers again for all the time and effort in helping us to improve our manuscript.

References:

1. Zhang H, Qian YL, Li C, Liu D, Wang L, Wang XY, Liu MJ, Liu H, Zhang S, Guo XY, Yang JX, Ding HL, Koo JW, Mouzon E, Deisseroth K, Nestler EJ, Zachariou V, Han MH, Cao JL. Brain-Derived Neurotrophic Factor in the Mesolimbic Reward Circuitry Mediates Nociception in Chronic Neuropathic Pain. *Biological psychiatry*. 2017;82(8):608-18. Epub 2017/04/10. doi: 10.1016/j.biopsych.2017.02.1180. PubMed PMID: 28390647; PMCID: PMC5788809.
2. Zhou W, Ye C, Wang H, Mao Y, Zhang W, Liu A, Yang CL, Li T, Hayashi L, Zhao W, Chen L, Liu Y, Tao W, Zhang Z. Sound induces analgesia through corticothalamic circuits. *Science (New York, NY)*. 2022;377(6602):198-204. Epub 2022/07/21. doi: 10.1126/science.abn4663. PubMed PMID: 35857536; PMCID: PMC9636983.
3. Zhou W, Jin Y, Meng Q, Zhu X, Bai T, Tian Y, Mao Y, Wang L, Xie W, Zhong H, Zhang N, Luo MH, Tao W, Wang H, Li J, Li J, Qiu BS, Zhou JN, Li X, Xu H, Wang K, Zhang X, Liu Y, Richter-Levin G, Xu L, Zhang Z. A neural circuit for comorbid depressive symptoms in chronic pain. *Nature neuroscience*. 2019;22(10):1649-58. Epub 2019/08/28. doi: 10.1038/s41593-019-0468-2. PubMed PMID: 31451801.
4. Wang K, Donnelly CR, Jiang C, Liao Y, Luo X, Tao X, Bang S, McGinnis A, Lee M, Hilton MJ, Ji RR. STING suppresses bone cancer pain via immune and neuronal modulation. *Nature communications*. 2021;12(1):4558. Epub 2021/07/29. doi: 10.1038/s41467-021-24867-2. PubMed PMID: 34315904; PMCID: PMC8316360 the company. This activity is not related to this study. R.R.J. and C.R.D. filed a patent on STING-related treatment for cancer pain: "COMPOSITIONS AND METHODS FOR THE TREATMENT OF PAIN" (PCT/US2021/028384). All other authors declare no competing interests.
5. Smith ML, Asada N, Malenka RC. Anterior cingulate inputs to nucleus accumbens control the social transfer of pain and analgesia. *Science (New York, NY)*. 2021;371(6525):153-9. Epub 2021/01/09. doi: 10.1126/science.abe3040. PubMed PMID: 33414216; PMCID: PMC7952019.
6. Li Z, Lu YF, Li CL, Wang Y, Sun W, He T, Chen XF, Wang XL, Chen J. Social interaction with a cagemate in pain facilitates subsequent spinal nociception via activation of the medial prefrontal cortex in rats. *Pain*. 2014;155(7):1253-61. Epub 2014/04/05. doi: 10.1016/j.pain.2014.03.019. PubMed PMID: 24699208.
7. Smith ML, Hostetler CM, Heinricher MM, Ryabinin AE. Social transfer of pain in mice. *Science advances*. 2016;2(10):e1600855. Epub 2016/10/25. doi: 10.1126/sciadv.1600855. PubMed PMID: 27774512; PMCID: PMC5072181.
8. Smith ML, Walcott AT, Heinricher MM, Ryabinin AE. Anterior Cingulate Cortex Contributes to Alcohol Withdrawal- Induced and Socially Transferred Hyperalgesia. *eNeuro*. 2017;4(4). Epub 2017/08/09. doi: 10.1523/eneuro.0087-17.2017. PubMed PMID: 28785727; PMCID: PMC5526654.
9. Lu YF, Ren B, Ling BF, Zhang J, Xu C, Li Z. Social interaction with a cagemate in pain increases allogrooming and induces pain hypersensitivity in the observer rats. *Neuroscience letters*. 2018;662:385-8. Epub 2017/11/06. doi: 10.1016/j.neulet.2017.10.063. PubMed PMID: 29102786.
10. Krishnan V, Han MH, Graham DL, Berton O, Renthal W, Russo SJ, Laplant Q, Graham A, Lutter M, Lagace DC, Ghose S, Reister R, Tannous P, Green TA, Neve RL, Chakravarty S, Kumar A, Eisch AJ, Self DW, Lee FS, Tamminga CA, Cooper DC, Gershenfeld HK, Nestler EJ. Molecular adaptations underlying susceptibility and resistance to social defeat in brain reward regions. *Cell*. 2007;131(2):391-404. Epub 2007/10/25. doi: 10.1016/j.cell.2007.09.018. PubMed PMID: 17956738.
11. Banissy MJ, Kanai R, Walsh V, Rees G. Inter-individual differences in empathy are reflected in

- human brain structure. *NeuroImage*. 2012;62(3):2034-9. Epub 2012/06/12. doi: 10.1016/j.neuroimage.2012.05.081. PubMed PMID: 22683384; PMCID: PMC3778747.
12. Guendelman S, Bayer M, Prehn K, Dziobek I. Regulating negative emotions of others reduces own stress: Neurobiological correlates and the role of individual differences in empathy. *NeuroImage*. 2022;254:119134. Epub 2022/03/31. doi: 10.1016/j.neuroimage.2022.119134. PubMed PMID: 35351648.
 13. McDonnell A, Schulman B, Ali Z, Dib-Hajj SD, Brock F, Cobain S, Mainka T, Vollert J, Tarabar S, Waxman SG. Inherited erythromelalgia due to mutations in SCN9A: natural history, clinical phenotype and somatosensory profile. *Brain : a journal of neurology*. 2016;139(Pt 4):1052-65. Epub 2016/02/28. doi: 10.1093/brain/aww007. PubMed PMID: 26920677.
 14. Geha P, Yang Y, Estacion M, Schulman BR, Tokuno H, Apkarian AV, Dib-Hajj SD, Waxman SG. Pharmacotherapy for Pain in a Family With Inherited Erythromelalgia Guided by Genomic Analysis and Functional Profiling. *JAMA neurology*. 2016;73(6):659-67. Epub 2016/04/19. doi: 10.1001/jamaneurol.2016.0389. PubMed PMID: 27088781.
 15. Mis MA, Yang Y, Tanaka BS, Gomis-Perez C, Liu S, Dib-Hajj F, Adi T, Garcia-Milian R, Schulman BR, Dib-Hajj SD, Waxman SG. Resilience to Pain: A Peripheral Component Identified Using Induced Pluripotent Stem Cells and Dynamic Clamp. *The Journal of neuroscience : the official journal of the Society for Neuroscience*. 2019;39(3):382-92. Epub 2018/11/22. doi: 10.1523/jneurosci.2433-18.2018. PubMed PMID: 30459225; PMCID: PMC6335750.
 16. Du X, Hao H, Gigout S, Huang D, Yang Y, Li L, Wang C, Sundt D, Jaffe DB, Zhang H, Gamper N. Control of somatic membrane potential in nociceptive neurons and its implications for peripheral nociceptive transmission. *Pain*. 2014;155(11):2306-22. Epub 2014/08/30. doi: 10.1016/j.pain.2014.08.025. PubMed PMID: 25168672; PMCID: PMC4247381.
 17. Passmore GM, Selyanko AA, Mistry M, Al-Qatari M, Marsh SJ, Matthews EA, Dickenson AH, Brown TA, Burbidge SA, Main M, Brown DA. KCNQ/M currents in sensory neurons: significance for pain therapy. *The Journal of neuroscience : the official journal of the Society for Neuroscience*. 2003;23(18):7227-36. Epub 2003/08/09. doi: 10.1523/jneurosci.23-18-07227.2003. PubMed PMID: 12904483; PMCID: PMC6740665.
 18. Friedman AK, Juarez B, Ku SM, Zhang H, Calizo RC, Walsh JJ, Chaudhury D, Zhang S, Hawkins A, Dietz DM, Murrough JW, Ribadeneira M, Wong EH, Neve RL, Han MH. KCNQ channel openers reverse depressive symptoms via an active resilience mechanism. *Nature communications*. 2016;7:11671. Epub 2016/05/25. doi: 10.1038/ncomms11671. PubMed PMID: 27216573; PMCID: PMC4890180.
 19. Zhang F, Liu S, Jin L, Tang L, Zhao X, Yang T, Wang Y, Huo B, Liu R, Li H. Antinociceptive efficacy of retigabine and flupirtine for gout arthritis pain. *Pharmacology*. 2020;105:471-6. doi: 10.1159/000505934. PubMed PMID: 32062659.
 20. Wang HR, Hu SW, Zhang S, Song Y, Wang XY, Wang L, Li YY, Yu YM, Liu H, Liu D, Ding HL, Cao JL. KCNQ channels in the mesolimbic reward circuit regulate nociception in chronic pain in mice. *Neurosci Bull*. 2021;37(5):597-610. Epub 2021/04/27. doi: 10.1007/s12264-021-00668-x. PubMed PMID: 33900570; PMCID: PMC8099961.
 21. Costi S, Morris LS, Kirkwood KA, Hoch M, Corniquel M, Vo-Le B, Iqbal T, Chadha N, Pizzagalli DA, Whitton A, Bevilacqua L, Jha MK, Ursu S, Swann AC, Collins KA, Salas R, Bagiella E, Parides MK, Stern ER, Iosifescu DV, Han MH, Mathew SJ, Murrough JW. Impact of the KCNQ2/3 Channel Opener Ezogabine on Reward Circuit Activity and Clinical Symptoms in Depression: Results From a

- Randomized Controlled Trial. *The American journal of psychiatry*. 2021;178(5):437-46. Epub 2021/03/04. doi: 10.1176/appi.ajp.2020.20050653. PubMed PMID: 33653118; PMCID: PMC8791195.
22. Costi S, Han MH, Murrrough JW. The Potential of KCNQ Potassium Channel Openers as Novel Antidepressants. *CNS drugs*. 2022;36(3):207-16. Epub 2022/03/09. doi: 10.1007/s40263-021-00885-y. PubMed PMID: 35258812.
 23. Tan A, Costi S, Morris LS, Van Dam NT, Kautz M, Whitton AE, Friedman AK, Collins KA, Ahle G, Chadha N, Do B, Pizzagalli DA, Iosifescu DV, Nestler EJ, Han MH, Murrrough JW. Effects of the KCNQ channel opener ezogabine on functional connectivity of the ventral striatum and clinical symptoms in patients with major depressive disorder. *Molecular psychiatry*. 2020;25(6):1323-33. Epub 2018/11/06. doi: 10.1038/s41380-018-0283-2. PubMed PMID: 30385872; PMCID: PMC6494706.
 24. Hemington KS, Rogachov A, Cheng JC, Bosma RL, Kim JA, Osborne NR, Inman RD, Davis KD. Patients with chronic pain exhibit a complex relationship triad between pain, resilience, and within- and cross-network functional connectivity of the default mode network. *Pain*. 2018;159(8):1621-30. Epub 2018/04/27. doi: 10.1097/j.pain.0000000000001252. PubMed PMID: 29697536.
 25. You B, Jackson T. Gray Matter Volume Differences Between More Versus Less Resilient Adults with Chronic Musculoskeletal Pain: A Voxel-based Morphology Study. *Neuroscience*. 2021;457:155-64. Epub 2021/01/24. doi: 10.1016/j.neuroscience.2021.01.019. PubMed PMID: 33484820.
 26. Li F, Jackson T. Gray matter volume differences between lower, average, and higher pain resilience subgroups. *Psychophysiology*. 2020;57(10):e13631. Epub 2020/07/06. doi: 10.1111/psyp.13631. PubMed PMID: 32621781.
 27. Nair-Roberts RG, Chatelain-Badie SD, Benson E, White-Cooper H, Bolam JP, Ungless MA. Stereological estimates of dopaminergic, GABAergic and glutamatergic neurons in the ventral tegmental area, substantia nigra and retrorubral field in the rat. *Neuroscience*. 2008;152(4):1024-31. Epub 2008/03/22. doi: 10.1016/j.neuroscience.2008.01.046. PubMed PMID: 18355970; PMCID: PMC2575227.
 28. Yamaguchi T, Qi J, Wang HL, Zhang S, Morales M. Glutamatergic and dopaminergic neurons in the mouse ventral tegmental area. *The European journal of neuroscience*. 2015;41(6):760-72. Epub 2015/01/13. doi: 10.1111/ejn.12818. PubMed PMID: 25572002; PMCID: PMC4363208.
 29. Warlow SM, Singhal SM, Hollon NG, Faget L, Dowlat DS, Zell V, Hunker AC, Zweifel LS, Hnasko TS. Mesoaccumbal glutamate neurons drive reward via glutamate release but aversion via dopamine co-release. *Neuron*. 2024;112(3):488-99.e5. Epub 2023/12/13. doi: 10.1016/j.neuron.2023.11.002. PubMed PMID: 38086374.
 30. Mu L, Liu X, Yu H, Vickstrom CR, Friedman V, Kelly TJ, Hu Y, Su W, Liu S, Mantsch JR, Liu QS. cAMP-mediated upregulation of HCN channels in VTA dopamine neurons promotes cocaine reinforcement. *Molecular psychiatry*. 2023;28(9):3930-42. Epub 2023/10/17. doi: 10.1038/s41380-023-02290-x. PubMed PMID: 37845497; PMCID: PMC10730389.
 31. Juarez B, Morel C, Ku SM, Liu Y, Zhang H, Montgomery S, Gregoire H, Ribeiro E, Crumiller M, Roman-Ortiz C, Walsh JJ, Jackson K, Croote DE, Zhu Y, Zhang S, Vendruscolo LF, Edwards S, Roberts A, Hodes GE, Lu Y, Calipari ES, Chaudhury D, Friedman AK, Han MH. Midbrain circuit regulation of individual alcohol drinking behaviors in mice. *Nature communications*. 2017;8(1):2220. Epub 2017/12/22. doi: 10.1038/s41467-017-02365-8. PubMed PMID: 29263389; PMCID: PMC5738419.
 32. Morel C, Montgomery S, Han MH. Nicotine and alcohol: the role of midbrain dopaminergic neurons in drug reinforcement. *The European journal of neuroscience*. 2019;50(3):2180-200. Epub

- 2018/09/27. doi: 10.1111/ejn.14160. PubMed PMID: 30251377; PMCID: PMC6431587.
33. Chaudhury D, Walsh JJ, Friedman AK, Juarez B, Ku SM, Koo JW, Ferguson D, Tsai HC, Pomeranz L, Christoffel DJ, Nectow AR, Ekstrand M, Domingos A, Mazei-Robison MS, Mouzon E, Lobo MK, Neve RL, Friedman JM, Russo SJ, Deisseroth K, Nestler EJ, Han MH. Rapid regulation of depression-related behaviours by control of midbrain dopamine neurons. *Nature*. 2013;493(7433):532-6. Epub 2012/12/14. doi: 10.1038/nature11713. PubMed PMID: 23235832; PMCID: PMC3554860.
34. Friedman AK, Walsh JJ, Juarez B, Ku SM, Chaudhury D, Wang J, Li X, Dietz DM, Pan N, Vialou VF, Neve RL, Yue Z, Han MH. Enhancing depression mechanisms in midbrain dopamine neurons achieves homeostatic resilience. *Science (New York, NY)*. 2014;344(6181):313-9. Epub 2014/04/20. doi: 10.1126/science.1249240. PubMed PMID: 24744379; PMCID: PMC4334447.
35. Holz NE, Tost H, Meyer-Lindenberg A. Resilience and the brain: a key role for regulatory circuits linked to social stress and support. *Molecular psychiatry*. 2020;25(2):379-96. Epub 2019/10/20. doi: 10.1038/s41380-019-0551-9. PubMed PMID: 31628419.
36. Zhai X, Zhou D, Han Y, Han MH, Zhang H. Noradrenergic modulation of stress resilience. *Pharmacological research*. 2023;187:106598. Epub 2022/12/09. doi: 10.1016/j.phrs.2022.106598. PubMed PMID: 36481260.
37. Morales M, Margolis EB. Ventral tegmental area: cellular heterogeneity, connectivity and behaviour. *Nature reviews Neuroscience*. 2017;18(2):73-85. Epub 2017/01/06. doi: 10.1038/nrn.2016.165. PubMed PMID: 28053327.
38. Faget L, Osakada F, Duan J, Ressler R, Johnson AB, Proudfoot JA, Yoo JH, Callaway EM, Hnasko TS. Afferent Inputs to Neurotransmitter-Defined Cell Types in the Ventral Tegmental Area. *Cell Rep*. 2016;15(12):2796-808. Epub 2016/06/09. doi: 10.1016/j.celrep.2016.05.057. PubMed PMID: 27292633; PMCID: PMC4917450.
39. Hnasko TS, Hjelmstad GO, Fields HL, Edwards RH. Ventral tegmental area glutamate neurons: electrophysiological properties and projections. *The Journal of neuroscience : the official journal of the Society for Neuroscience*. 2012;32(43):15076-85. Epub 2012/10/27. doi: 10.1523/jneurosci.3128-12.2012. PubMed PMID: 23100428; PMCID: PMC3685320.
40. Taylor SR, Badurek S, Dileone RJ, Nashmi R, Minichiello L, Picciotto MR. GABAergic and glutamatergic efferents of the mouse ventral tegmental area. *J Comp Neurol*. 2014;522(14):3308-34. doi: 10.1002/cne.23603. PubMed PMID: 24715505; PMCID: PMC4107038.
41. Sun H, Li Z, Qiu Z, Shen Y, Guo Q, Hu SW, Ding HL, An S, Cao JL. A common neuronal ensemble in nucleus accumbens regulates pain-like behaviour and sleep. *Nature communications*. 2023;14(1):4700. Epub 2023/08/06. doi: 10.1038/s41467-023-40450-3. PubMed PMID: 37543693; PMCID: PMC10404280.
42. Sandoval-Rodríguez R, Parra-Reyes JA, Han W, Rueda-Orozco PE, Perez IO, de Araujo IE, Tellez LA. D1 and D2 neurons in the nucleus accumbens enable positive and negative control over sugar intake in mice. *Cell reports*. 2023;42(3):112190. Epub 2023/03/02. doi: 10.1016/j.celrep.2023.112190. PubMed PMID: 36857179; PMCID: PMC10154129.
43. Francis TC, Chandra R, Friend DM, Finkel E, Dayrit G, Miranda J, Brooks JM, Iñiguez SD, O'Donnell P, Kravitz A, Lobo MK. Nucleus accumbens medium spiny neuron subtypes mediate depression-related outcomes to social defeat stress. *Biological psychiatry*. 2015;77(3):212-22. Epub 2014/09/01. doi: 10.1016/j.biopsych.2014.07.021. PubMed PMID: 25173629; PMCID: PMC5534173.

44. Brinschwitz K, Dittgen A, Madai VI, Lommel R, Geisler S, Veh RW. Glutamatergic axons from the lateral habenula mainly terminate on GABAergic neurons of the ventral midbrain. *Neuroscience*. 2010;168(2):463-76. Epub 2010/04/01. doi: 10.1016/j.neuroscience.2010.03.050. PubMed PMID: 20353812.
45. Zhang L, Hernández VS, Swinny JD, Verma AK, Giesecke T, Emery AC, Mutig K, Garcia-Segura LM, Eiden LE. A GABAergic cell type in the lateral habenula links hypothalamic homeostatic and midbrain motivation circuits with sex steroid signaling. *Translational psychiatry*. 2018;8(1):50. Epub 2018/02/27. doi: 10.1038/s41398-018-0099-5. PubMed PMID: 29479060; PMCID: PMC5865187.
46. Hu H, Cui Y, Yang Y. Circuits and functions of the lateral habenula in health and in disease. *Nature reviews Neuroscience*. 2020;21(5):277-95. Epub 2020/04/10. doi: 10.1038/s41583-020-0292-4. PubMed PMID: 32269316.

REVIEWERS' COMMENTS

Reviewer #1 (Remarks to the Author):

Following the revisions implemented and rearranged in this manuscript entitled "Midbrain glutamatergic circuit mechanism of resilience to pain in male mice" (NCOMMS-23-36972A), the manuscript has evolved into a notably compelling and engaging piece. I find the latest version to be exceptionally well-crafted and have no further substantive concerns. In my assessment, I deem this manuscript not only acceptable but highly commendable for publication in your esteemed scientific journal, Nature Communications.

Reviewer #3 (Remarks to the Author):

The authors have revised the manuscript rigorously, which includes additional experiments. All the questions and comments in my previous review, and in my opinion also the other reviewer's comments, have been appropriately addressed. I believe that the manuscript is now ready to be published in Nature Communications.

Reviewer #4 (Remarks to the Author):

The authors have addressed most of my concerns, and the manuscript has been substantially improved. No more questions.

Point-to-point response to reviewer's comments

REVIEWER COMMENTS

Reviewer #1 (Remarks to the Author):

Following the revisions implemented and rearranged in this manuscript entitled "Midbrain glutamatergic circuit mechanism of resilience to pain in male mice" (NCOMMS-23-36972A), the manuscript has evolved into a notably compelling and engaging piece. I find the latest version to be exceptionally well-crafted and have no further substantive concerns. In my assessment, I deem this manuscript not only acceptable but highly commendable for publication in your esteemed scientific journal, Nature Communications.

We are thrilled to receive such a glowing endorsement of our revised manuscript from you. Your assessment that the manuscript has evolved into a "notably compelling and engaging piece" is truly encouraging. Your positive evaluation provides us with great confidence as we move closer to sharing our findings with the wider scientific community. We sincerely thank you for your meticulous review, insightful feedback, and unwavering support throughout the revision process.

Reviewer #3 (Remarks to the Author):

The authors have revised the manuscript rigorously, which includes additional experiments. All the questions and comments in my previous review, and in my opinion also the other reviewer's comments, have been appropriately addressed. I believe that the manuscript is now ready to be published in Nature Communications.

We are delighted that you now consider our manuscript ready for publication in Nature Communications. Your expert guidance and constructive criticism have been invaluable in shaping our research into its current form, and we are sincerely grateful for your time and dedication throughout the review process.

Reviewer #4 (Remarks to the Author):

The authors have addressed most of my concerns, and the manuscript has been substantially improved. No more questions.

We are pleased to learn that our revisions have successfully addressed the majority of your concerns, leading to a substantial improvement in our manuscript. We are grateful for your thorough review, constructive feedback, and the opportunity to refine our work based on your insights. Thank you once again for your valuable input and for supporting the advancement of our research.